



# Observations and simulation of intense convection embedded in a warm conveyor belt – how ambient vertical wind shear determines the dynamical impact

Annika Oertel[1,2], Michael Sprenger[1], Hanna Joos[1], Maxi Boettcher[1], Heike Konow[3], Martin Hagen[4], and Heini Wernli[1]

[1]Institute for Atmospheric and Climate Science, ETH Zürich, Zürich, Switzerland
[2]Institute for Meteorology and Climate Research, Karlsruhe Institute of Technology (KIT), Karlsruhe, Germany
[3]Meteorological Institute, University of Hamburg, Hamburg, Germany
[4]Deutsches Zentrum für Luft- und Raumfahrt, Institut für Physik der Atmosphäre, Oberpfaffenhofen, Germany

**Correspondence:** Annika Oertel (annika.oertel@kit.edu)

**Abstract.** Warm conveyor belts (WCBs) are dynamically important, strongly ascending and mostly stratiform cloud-forming airstreams in extratropical cyclones. Despite the predominantly stratiform character of the WCB's large-scale cloud band, convective clouds can be embedded in it. This embedded convection leads to a heterogeneously structured cloud band with locally enhanced hydrometeor content, intense surface precipitation and substantial amounts of graupel in the middle tro-

poshere. Recent studies showed that embedded convection forms dynamically relevant quasi-horizontal potential vorticity (PV) dipoles in the upper troposphere. Thereby one pole can reach strongly negative PV values associated with inertial or symmetric instability near the upper-level PV waveguide, where it can interact with and modify the upper-level jet. This study analyses the characteristics of embedded convection in the WCB of cyclone Sanchez based on WCB online trajectories from a convection-permitting simulation and airborne radar observations during the North Atlantic Waveguide and Downstream

Impact EXperiment (NAWDEX) field campaign (IOPs 10 and 11). In the first part, we present the radar reflectivity structure of the WCB and corroborate its heterogeneous cloud structure and the occurrence of embedded convection. Radar observations in three different sub-regions of the WCB cloud band reveal the differing intensity of its embedded convection, which is qualitatively confirmed by the ascent rates of the online WCB trajectories. The detailed ascent behaviour of the WCB trajectories reveals that very intense convection with ascent rates of 600 hPa in 30-60 min occurs, in addition to comparatively moderate

convection with slower ascent velocities as reported in previous case studies. In the second part of this study, a systematic Lagrangian composite analysis based on online trajectories for two sub-categories of WCB-embedded convection – moderate and intense convection – is performed. Composites of the cloud and precipitation structure confirm the large influence of embedded convection: Intense convection produces locally very intense surface precipitation with peak values exceeding 6 mm in 15 minutes and large amounts of graupel of up to $2.8\,\mathrm{g\,kg^{-1}}$ in the middle troposphere (compared to 3.9 mm and $1.0\,\mathrm{g\,kg^{-2}}$ for

the moderate convective WCB sub-category). In the upper troposphere, both convective WCB trajectory sub-categories form a small-scale and weak PV dipole, with one pole reaching weakly negative PV values. However, for this WCB case study – in contrast to previous case studies reporting convective PV dipoles in the WCB ascent region with the negative PV pole near




the upper-level jet – the negative PV pole is located east of the convective ascent region, i.e., away from the upper-level jet. Moreover, the PV dipole formed by the intense convective WCB trajectories is weaker and has a smaller horizontal and vertical extent compared to a previous NAWDEX case study of WCB-embedded convection, despite faster ascent rates in this case. The absence of a strong upper-level jet and the weak vertical shear of the ambient wind in cyclone Sanchez are accountable for the weak diabatic PV modification in the upper troposphere. This implies that the strength of embedded convection alone is not a reliable measure for the effect of embedded convection on upper-level PV modification and its impact on the upper-level jet. Instead, the profile of vertical wind shear and the alignment of embedded convection with a strong upper-level jet play a key role for the formation of coherent negative PV features near the jet. Finally, these results highlight the large case-to-case variability of embedded convection not only in terms of frequency and intensity of embedded convection in WCBs but also in terms of its dynamical implications.

## 1  Introduction

Convective storms form an important part of the climate system (Houze, 1973) and still constitute one of the major obstacles for the improvement of weather and climate predictions (e.g., Ban et al., 2014; Sherwood et al., 2014; Holloway et al., 2014). In contrast to the well-studied so-called isolated and often tropical convection, convective clouds in the mid-latitudes can be an integral part of larger-scale weather systems. In the extratropical storm track region, convective clouds can be embedded within larger-scale cloud bands of extratropical cyclones (e.g., Browning et al., 1973; Browning and Roberts, 1999; Neiman et al., 1993; Naud et al., 2015; Flaounas et al., 2016), resulting in mesoscale variability of the cloud and precipitation structure. Recent studies also corroborated the occurrence of convection embedded in the warm conveyor belt (WCB) airstream (Binder, 2016; Crespo and Posselt, 2016; Rasp et al., 2016; Flaounas et al., 2016, 2018; Oertel et al., 2019, 2020; Blanchard et al., 2020) as originally proposed by Neiman et al. (1993) in their so-called 'escalator-elevator' concept. In the classical perspective, the WCB (e.g. Browning, 1990; Madonna et al., 2014) is described as a gradually ascending and mostly stratiform-cloud-forming airstream in extratropical cyclones. Its slantwise ascending motion has been referred to as 'escalator'-like ascent. In contrast, the phases of rapid convective ascent embedded within this large-scale airstream were referred to as the 'elevator'.

The convective activity can be directly embedded within the large-scale WCB cloud band and is characterized by a locally denser cloud with enhanced hydrometeor content and enhanced surface precipitation intensity (Oertel et al., 2019, 2020). From an observational perspective, WCB-embedded convection is typically 'hidden' within the large-scale cloud band of the WCB. In radar observations it has been identified through increased horizontal heterogeneity of the radar reflectivity, the absence of a well-defined bright band and the occurrence of narrow plumes of enhanced radar reflectivity (Neiman et al., 1993; Crespo and Posselt, 2016; Oertel et al., 2019). Compared to deep convective storms, where reflectivities exceed 40-50 dB$Z$ in the upper troposphere and updrafts exceed 10-15 m s$^{-1}$ (e.g., Carbone, 1982; Miller et al., 1988; Steiner et al., 1995), the ascent within embedded convection was reported to be slower with 1-5 m s$^{-1}$ with radar reflectivities reaching approximately 20-30 dB$Z$ (Crespo and Posselt, 2016; Oertel et al., 2019; Binder et al., 2020; Gehring et al., 2020; Blanchard et al., 2020).



It is well known that WCBs are dynamically relevant airstreams in extratropical cyclones that can modify the larger-scale flow evolution during their ascent from the boundary layer into the upper troposphere (e.g., Grams et al., 2011; Binder et al., 2016; Joos and Forbes, 2016; Rodwell et al., 2017). Their impact on the upper-tropospheric flow can be conveniently analysed with the potential vorticity (PV) framework (Hoskins et al., 1985) because the intense cloud diabatic processes during their deep ascent distinctly modify the PV distribution, typically leading to a wide region of low-PV air in the upper tropospheric

ridge (e.g., Wernli and Davies, 1997; Pomroy and Thorpe, 2000; Madonna et al., 2014; Methven, 2015). Harvey et al. (2020) and Oertel et al. (2020) showed that smaller-scale regions of strongly enhanced diabatic heating, such as from convective ascent, embedded in this large-scale WCB ascent region can form quasi-horizontal upper-level PV dipoles with a positive PV pole to the east towards the ridge, and a region of negative PV to the west near the upper-level PV waveguide. The latter can be relevant for the upper-level jet structure and the larger-scale flow evolution. The smaller-scale and narrow regions of increased

hydrometeor production associated with the locally confined regions of embedded convective ascent lead to strong horizontal diabatic heating gradients ($\nabla_h \dot{\theta}$), which are substantially stronger than for a purely slantwise ascending WCB that forms an extended region of moderate heating. This subsequently influences the resulting PV signature (Hoskins et al., 1985)

$$\frac{D}{Dt}PV = \frac{1}{\rho}\boldsymbol{\omega} \cdot \nabla\dot{\theta} = \frac{1}{\rho}\left[(f+\zeta)\frac{\partial\dot{\theta}}{\partial z} + \boldsymbol{\omega}_h \cdot \nabla_h\dot{\theta}\right] \tag{1}$$

where PV is defined as (Ertel, 1942)

$$PV = \frac{1}{\rho}\boldsymbol{\omega} \cdot \nabla\theta \tag{2}$$

and $\rho$ is density, $\theta$ is potential temperature, $\dot{\theta}$ represents latent heating or cooling, and $\boldsymbol{\omega}$ is 3D absolute vorticity defined as

$$\boldsymbol{\omega} = \nabla \times \mathbf{u} + 2\boldsymbol{\Omega} = \xi\mathbf{i} + \eta\mathbf{j} + (f+\zeta)\mathbf{k}. \tag{3}$$

Thereby, $\mathbf{u}$ is the 3D wind vector, $\boldsymbol{\Omega}$ is the vector of earth rotation, $\xi$ and $\eta$ are the horizontal vorticity components in x- and y-direction, $f$ is the Coriolis parameter and $f+\zeta$ is the absolute vertical vorticity and $\boldsymbol{\omega}_h$ denotes the horizontal vorticity

($\boldsymbol{\omega}_h = \xi\mathbf{i} + \eta\mathbf{j}$).

Scale analysis shows that the vertical components of Eq. 1 (first term) dominante on large scales (such as WCB ascent with length scales of the order of $10^6$ m, where $Ro \ll 1$, with the Rossby number $Ro = \frac{U}{f \cdot L}$), whereas for smaller length scales such as convection ($Ro > 1$) the horizontal components (second term in Eq. 1) become increasingly important (Haynes and McIntyre, 1987; Martínez-Alvarado et al., 2016; Harvey et al., 2020). The latter also applies if the length scale of diabatic

heating occurs on a smaller spatial scale than the large-scale flow, as e.g., for localised heating embedded in the large-scale WCB ascent (Harvey et al., 2020). Idealized simulations (Chagnon and Gray, 2009) show a gradual tilt from a vertically-oriented PV dipole – where $(f+\zeta)\frac{\partial\dot{\theta}}{\partial z}$ dominates – to a horizontally-oriented PV dipole – where $\boldsymbol{\omega}_h \cdot \nabla_h\dot{\theta}$ becomes increasingly important – with a decrease of the scale of the system and an increase of the environmental wind shear.





As a consequence, the typical large-scale and slantwise WCB ascent forms predominantly a vertical PV dipole (Eq. 1, first
term) with increased low-level PV and a wide-spread region of low-PV air in the upper troposphere (e.g., Wernli and Davies,
1997; Joos and Wernli, 2012; Madonna et al., 2014). In contrast, convection in a vertically sheared environment forms quasi-
horizontal mid- to upper-level PV dipoles that are centered around the convective updraft with the negative PV pole to the left
of the vertical wind shear vector and the positive PV pole to the right of the vertical wind shear vector (Eq. 1, second term).
The formation of such quasi-horizontal PV dipole structures have been found in idealized simulations of isolated cumulus-
scale convection (Chagnon and Gray, 2009), in case studies of mesoscale convective systems (Davis and Weisman, 1994;
Chagnon and Gray, 2009; Hitchman and Rowe, 2017; Clarke et al., 2019), in mid-latitude convective updrafts with varying
large-scale flow conditions (Weijenborg et al., 2015, 2017; Müller et al., 2020), and embedded in the baroclinic WCB ascent
region (Harvey et al., 2020; Oertel et al., 2020). These PV anomalies can reach a magnitude of $\pm 10 \, \mathrm{PVU}$, and hence, exceed
the typical range of synoptic-scale PV.

The convective-scale PV anomalies are dynamically relevant as they can interact with the larger-scale flow (Clarke et al.,
2019; Harvey et al., 2020; Oertel et al., 2020). Oertel et al. (2020) and Harvey et al. (2020) showed that convective activity and
narrow bands of strong latent heating embedded in the baroclinic region ahead of an upper-level trough can form elongated
bands of negative PV at the jet-facing side of the heating region. These negative PV bands strengthen the isentropic PV
gradient across the tropopause and are associated with an accelerated upper-level jet. Hence, in certain synoptic situations, the
dynamically unstable[1] region of convectively formed negative PV and its associated anticyclonic absolute vertical vorticity can
become relevant as it interacts with and modifies the upper-level PV waveguide (Harvey et al., 2020; Oertel et al., 2020).

Although recent case studies have corroborated the presence of embedded convection in WCBs (Rasp et al., 2016; Crespo
and Posselt, 2016; Flaounas et al., 2016, 2018; Oertel et al., 2019), several major issues are still uncertain: (i) Where and how
often is convection generally embedded in the WCB ascent region? (ii) How variable is the strength of embedded convection
in WCBs? (iii) Does WCB-embedded convection consistently form coherent PV dipoles with the negative pole in vicinity to
the upper-level jet (as shown in the previous case study by Oertel et al., 2020)?

While the first point requires a climatological analysis of embedded convection, here we specifically address the variable
strength and dynamical signatures of embedded convection in a specific WCB. The aim of this detailed WCB case study
is two-fold: On the one hand, we corroborate the concept of embedded convection in WCBs and identify WCB-embedded
convection with differing strengths based on rare airborne radar observations of a WCB and online trajectories in a convection-
permitting simulation. On the other hand, we systematically analyse the characteristics and the PV signature associated with (i)
intense and (ii) moderate WCB-embedded convection. Therefore, we investigate the WCB associated with the surface cyclone
and upper-level PV cutoff Sanchez that occurred during the *North Atlantic Waveguide and Downstream Impact EXperiment*
(NAWDEX, Schäfler et al., 2018) field campaign.

Specifically, we address the following questions:

---

[1]Negative PV values in the northern hemisphere are related to either hydrostatic, inertial or symmetric instability (Hoskins, 2015, e.g.,), and have been
associated with mesoscale circulations (e.g., Bennetts and Hoskins, 1979; Schultz and Schumacher, 1999; Volonté et al., 2020).



1. What do airborne radar observations of the WCB cloud band in cyclone Sanchez reveal about the strength of embedded convective activity (Sect. 4)?

2. What are (thermodynamic) characteristics associated with WCB-embedded convection in this case study (Sect. 5.3 and 5.4)?

3. How important is the strength of WCB-embedded convection for the resulting upper-level PV modification compared to characteristics of the ambient flow (Sect. 5.5 and 5.6)?

This study is structured in the following way: After introducing the data and methodology (Sect. 2) and presenting the evolution of the WCB and its embedded convection (Sect. 3), the first part of the results (Sect. 4) shows the radar-based perspective of the WCB cloud band and qualitatively compares the observations with online trajectories from a convection-permitting simulation. In the second part (Sect. 5), we perform a 3D Lagrangian composite analysis to investigate the PV modification by two categories of convective WCB trajectories – intensely and more moderately ascending convective WCB trajectories. Throughout this study, results are compared with cyclone Vladiana, a previous case study of WCB-embedded convection, which also occurred over the North Atlantic during the NAWDEX field campaign two weeks earlier (Oertel et al., 2020).

## 2 Data

### 2.1 Airborne radar observations

The investigated cyclone Sanchez occurred in October 2016 during the NAWDEX field campaign (IOPs 10 and 11, Schäfler et al., 2018). During this campaign the German research aircraft HALO (High Altitude and LOng Range Research Aircraft) observed the cloud structure of the cyclone and its associated WCB twice on two consecutive days. The research aircraft carries the HALO Microwave Package (HAMP), which includes the high-sensitivity cloud radar MIRA-36 (Melchiona et al., 2008; Mech et al., 2014). The downward pointing Doppler radar operates at a frequency of $35.5\,\mathrm{GHz}$ and reaches an expected airborne sensitivity of $-38\,\mathrm{dB}Z$ at $5\,\mathrm{km}$ distance (Mech et al., 2014), which reduces to $\approx -30\,\mathrm{dB}Z$ at $13\,\mathrm{km}$ distance (Konow et al., 2019). The measured along-flight radar reflectivity $Z$ has been post-processed, offset-corrected and corrected for aircraft attitude (Ewald et al., 2019; Konow et al., 2019) and is obtained from Konow et al. (2018).

To characterize the large-scale cloud band of the WCB and its embedded convection we analyse the radar reflectivity structure where the HALO flight track intersects the WCB cloud band. The location of WCB air parcels is determined by the operational analyses from the ECMWF (see Sect. 2.2). The radar reflectivity structure of convective clouds is relatively heterogeneous compared to homogeneous stratiform clouds (e.g., Steiner et al., 1995) and often appears as confined towers (Hogan et al., 2002; Murphy et al., 2017). Here, we visually identify convective regions, where narrow plumes of increased values of $Z$ rise above the $0\,^{\circ}\mathrm{C}$ isotherm (cf. Hogan et al., 2002; Murphy et al., 2017; Oertel et al., 2019). Further applied indications for convective activity are locally enhanced radar reflectivity, a broken cloud top, spatially heterogeneous cloud top height, and the absence of a pronounced radar bright band.



## 2.2 Offline WCB trajectories from ECMWF analyses

The large-scale WCB ascent region is identified with offline trajectories from the operational analyses from the *Integrated Forecasting System* (IFS) of the *European Centre for Medium-Range Weather Forecasts* (ECMWF), which is run with a spatial resolution of approximately 9 km with 137 vertical hybrid pressure-sigma levels [operational model version in October 2016 (cycl 41r2, ECMWF, 2016)] and includes parameterized shallow and deep convection (Tiedtke, 1989; Bechtold et al., 2008, 2014). The trajectories are computed with the Lagrangian analysis tool LAGRANTO (Wernli and Davies, 1997; Sprenger and Wernli, 2015) from 1-hourly 3D wind fields that are interpolated to a regular 0.5° grid and combine 3-hourly analyses with operational short-term forecasts in between. To identify the WCB, trajectories were started globally every hour and every 80 km in the horizontal on 14 vertical levels (every 25 hPa between 1025 and 700 hPa) similar to the method applied in Madonna et al. (2014). WCB trajectories are subsequently selected as trajectories with ascent rates of at least 600 hPa in 48 h (e.g., Madonna et al., 2014).

## 2.3 Convection-permitting simulation and online WCB trajectories

To simulate the WCB and its embedded convection, we use the fully elastic nonhydrostatic limited-area model COSMO (*COnsortium for Small-scale MOdeling*, Baldauf et al., 2011; Doms and Baldauf, 2018). The domain is centered in the eastern North Atlantic and extends from about 55°W to 5°E and 30°N to 65°N (e.g., Fig. 1j). The simulation is run in convection-permitting mode with a horizontal grid-spacing of 0.02° (∼2.2 km) and with 60 vertical levels. We employed a one-moment six-category cloud microphysics scheme including prognostic water vapour ($q_v$), liquid (LWC) and ice (IWC) cloud water content, rain (RWC), snow (SWC) and graupel (GWC), which is important for the explicit simulation of deep convection (Baldauf et al., 2011). While deep convection is treated explicitly at 2.2 km (e.g., Ban et al., 2014), we applied the reduced Tiedtke scheme for shallow convection (Tiedtke, 1989; Baldauf et al., 2011). 3D COSMO fields were output every 15 minutes, which allows capturing the large temporal and spatial variability of embedded convection. The simulation is initialized at 00 UTC 07 October 2016, and runs for 106 hours with initial and lateral boundary conditions from the ECMWF analyses with a horizontal resolution of 0.1° every 6 h.

The detailed WCB ascent behaviour is analysed with online trajectories computed during the COSMO simulation (Miltenberger et al., 2013, 2014), which are used to identify phases of embedded convective ascent in the WCB ascent region (Rasp et al., 2016; Oertel et al., 2019, 2020) The online trajectories are started every 2 h during the first 96 h of the simulation from a predefined starting region (which is determined based on the offline WCB trajectory positions) at seven vertical levels (250, 500, 750, 1000, 1500, 2000 and 2500 m a.s.l.). The local strength of WCB trajectory ascent is quantified through the centered 2-h net pressure change $\Delta p_{2h}(t) = p(t+1h) - p(t-1h)$ along the online WCB trajectories, which is calculated for all trajectories that exceed an ascent rate of 600 hPa in 48 h. WCB ascent phases have previously been considered as convective if $\Delta p_{2h} < -320\,\mathrm{hPa}\,(2\,\mathrm{h})^{-1}$ (Rasp et al., 2016; Oertel et al., 2019). More extreme values of $\Delta p_{2h} < -600\,\mathrm{hPa}\,(2\,\mathrm{h})^{-1}$ can, however, also be embedded in the WCB ascent region (Rasp et al., 2016; Oertel et al., 2019, 2020).





## 3 Evolution of the WCB and its embedded convection

### 3.1 Synoptic situation

The surface cyclone that was later associated with the upper-level PV cutoff Sanchez has formed several days before the research flights on 09 and 10 October 2016. The genesis region south of 40°N is situated in a warm and moist environment where the equivalent potential temperature ($\theta_e$) at 850 hPa exceeds 335 K and is located south of the stronger, relatively zonal polar jet with 60 m s$^{-1}$ at 60°N. In the early phase, the surface cyclone remains quasi-stationary below an upper-level PV anomaly and west of a surface anticyclone and only travels slowly eastward and northward across the North Atlantic (Oertel, 2019, Fig. 5.2).

Eventually, on 08 October the surface cyclone propagates poleward influenced by the steering flow of the anticyclone located to its east and the upper-level trough that approaches the cyclone from the northwest (Fig. 1a,d). During the poleward propagation the surface cyclone intensifies and the cold and warm fronts strengthen in hand with the intensification of the cyclone (Fig. 1b,c). A first PV streamer breakup occurs early on 09 October (cf. Schäfler et al., 2018), but the stratospheric PV cutoff does not interact with the surface cyclone and propagates equatorward. The key interaction between the surface cyclone and an upper-level PV cutoff, which is clearly visible at 320 K, occurs later on 09 October at 45°N during the second PV streamer roll-up and cutoff formation. Then, the PV cutoff and the surface cyclone phase lock, leading to rapid intensification of the surface cyclone (Fig. 1b,c). During this intensification the surface pressure drops by 15 hPa within 15 h to a minimum of 985 hPa. The surface cyclone and the upper-level PV cutoff align barotropically (Fig. 1b,c) and propagate eastward towards southern Europe (not shown), where the PV cutoff and associated WCB lead to high-impact weather across southern France, including heavy precipitation, flooding and strong winds (Schäfler et al., 2018; Binder et al., submitted). Schäfler et al. (2018) and Oertel (2019, Chapter 5) provide a more detailed description of the synoptic situation and the evolution of the surface cyclone and associated upper-level PV cutoff. During this period (08-11 October) the forecast quality in the North Atlantic region was relatively low with a comparatively large root mean square error and ensemble spread of 500 hPa geopotential (Schindler et al., 2020). Uncertainty in the PV cutoff position was associated with a misrepresentation of the heavy precipitation event that occurred downstream over southern France (Binder et al., submitted).







**Figure 1.** Overview of the surface cyclone and PV cutoff Sanchez: **(a-c)** sea level pressure (SLP, grey contours, every 5 hPa), 2 PVU at 320 K (red) and equivalent potential temperature ($\theta_e$) at 850 hPa (colors, in K) from the ECMWF dataset, and lightning observations (yellow) at (from left to right) 21 UTC 08 October 2016, 21 UTC 09 October 2016, and 06 UTC 10 October 2016; **(d-f)** as **(a-c)** but for satellite-derived cloud top pressure (colors, in hPa); data from Meteosat Second Generation Satellites (EUMETSAT; Schmetz et al., 2002); **(g-i)** as **(a-c)** but for WCB trajectories (colored according to pressure) from the ECMWF dataset started 18 hours before the times shown in **(a-c)** and WCB trajectory positions (circles colored according to pressure) for the times shown in **(a-c)**; **(j-l)** centered 2-h pressure change ($\Delta p_{2h}$, colored circles, in hPa) along ascending WCB trajectories from the convection-resolving COSMO simulation, 2 PVU (red) contour and jet (colors, in m s$^{-1}$) at 320 K, and SLP (grey contours, every 5 hPa) from the COSMO simulation for the same times as shown above.

## 3.2 WCB ascent and cloud structure

During its lifecycle, the cyclone is associated with a prominent WCB (Fig. 1g-i). After the interaction with the upper-level trough and developing PV streamer at approximately 45°N, the WCB trajectories ascend poleward and form a pronounced anticyclonic branch that reaches 60°N (Fig. 1g,h). The WCB ascends primarily in two regions, in a narrow band ahead of the surface cold front and above the warm frontal surface, and forms a pronounced cloud band with cloud tops exceeding 200 hPa (Fig. 1d-f). The large-scale cloud band strongly associated with warm frontal WCB ascent covers large parts of the upper-level ridge (Fig. 1d,e) and wraps cyclonically around the cyclone center during the PV cutoff formation (Fig. 1e). Both cold and warm frontal cloud bands are characterized by high clouds with heterogeneously structured cloud tops (Fig. 1d-f). In particular, the cold frontal clouds form a narrow vertically extended band with cloud tops exceeding 150-200 hPa (Fig. 1e,f). South of 40°N, deep clouds with very high cloud tops coincide with lightning observations from the *World Wide Lightning Location Network* (WWLLN, Abarca et al., 2010) indicating that deep convective clouds are embedded in the extended WCB cloud band (Fig. 1a,b). This convective character of the cloud band is particularly pronounced in the early phase of the WCB (e.g., Fig. 1d).

## 3.3 Evolution of embedded convection

The position of the upper-level trough (cf. Fig. 1a,j), its thinning, and the PV streamer development (cf. Fig. 1b,k) as well as the PV cutoff formation are well captured by the COSMO simulation (cf. Fig. 1c,l). Moreover, the WCB ascent region of the COSMO simulation (Fig. 1j-l) agrees with the WCB ascent region of the ECMWF WCB trajectories (Fig. 1g-i) and coincides with the large-scale cloud band (Fig. 1d-f).

To investigate the WCB ascent behaviour, the 2-h ascent rates $\Delta p_{2h}$ along online trajectories are shown in Fig. 1j-l. Slantwise and stratiform cloud-forming WCB ascent is typically characterized by moderate $\Delta p_{2h}$ values of approximtely $-50$ to $-300$ hPa, while embedded convective ascent phases reach $\Delta p_{2h}$ values of $-320$ to $-400$ hPa and can even exceed $-600$ hPa (Rasp et al., 2016; Oertel et al., 2019). For cyclone Sanchez, such convective ascent phases are frequently embedded within the overall slantwise WCB ascent region (Fig. 1j-l). In agreement with the cloud structure and lightning observations (Fig. 1d), the most intense embedded convection ($\Delta p_{2h} < -600$ hPa) occurs frequently in the early phase before the PV cut-off formation near the cyclone center and at the cold front (Fig. 1j, magenta outline). After the northward propagation of the





cyclone, these intense convective ascent phases with $\Delta p_{2h} < -600$ hPa become scarcer, but $\Delta p_{2h}$ values still remain below $-400$ hPa (Fig. 1k,l).

During the PV streamer roll-up and cutoff formation and simultaneous deepening of the surface cyclone, the ascending WCB
air parcels aggregate mainly in two banded regions, ahead of the cold front and at the warm front, with $\Delta p_{2h}$ values that vary between $-25$ hPa and $-400$ to $-600$ hPa at the cold front (Fig. 1k,l). Convective ascent with $\Delta p_{2h} < -320$ hPa occurs ahead of the surface cold front and near the warm front northeast of the cyclone center and is directly embedded in the region of slower slantwise WCB ascent. Although convective ascent phases with $\Delta p_{2h} < -320$ hPa are embedded in the WCB ascent region at both the cold and warm front, the warm-frontal convective WCB ascent is generally more moderate with $\Delta p_{2h}$ values
of up to only $-350$ hPa, in contrast to the intense cold frontal convection with $\Delta p_{2h}$ values in the range of $-400$ to $-600$ hPa. In summary, convective activity identified from COSMO online trajectories is frequently embedded in the WCB of cyclone Sanchez at the cold and warm fronts. The embedded convective ascent phases show substantial variability in intensity, whereby the most intense convective ascent occurs near the cyclone center and ahead of the cold front primarily in the early phase and south of 45°N. This allows for an investigation of the effect of intense vs. moderate embedded convection on surface
precipitation and upper-level PV modification. This will follow in Sect. 5, after a detailed analysis of airborne radar observations in the next section.

## 4   Radar reflectivity structure of the WCB

During the research flights on 09 and 10 October, the HALO aircraft observed the WCB cloud band several times. The different flight legs captured different stages of the WCB, including slantwise and stratiform cloud-forming regions as well as two
convective regions with $\Delta p_{2h}$ values ranging from $-25$ hPa in the stratiform regions to $-400$ hPa in the convective regions near the flight track (Figs. 2a and 3a). As the southernmost flight legs of HALO were located at around 47°N, the most intense convective regions with $\Delta p_{2h} < -600$ hPa (Sect. 3.3 and Fig. 1j) were not captured. The differing WCB ascent behaviour seen in $\Delta p_{2h}$ values is also reflected in the radar observations. In the following, three contrasting radar reflectivity cross-sections of the extended WCB cloud band are discussed, whereby the large-scale WCB ascent region is identified through offline
trajectories from the ECMWF analyses.

The first observation of the WCB cloud band on 09 October took place from 13:00 to 14:00 UTC at around 48°N (Fig. 2a). At 13:00 UTC the HALO crossed the PV streamer from west to east and subsequently entered the warm sector across the warm front near the cyclone center (Fig. 2a). Before turning back and traversing the PV streamer a second time from east to west, thereby crossing the front, HALO observed the WCB cloud band between 13:20 and 14:00 UTC (Fig. 2a, yellow line). For the
second observation of the WCB cloud band on 09 October, HALO traversed the large-scale cloud band from west to east at 16:00 UTC north of 50°N (Fig. 2a, orange line).

*Stratiform cloud band related to slantwise WCB ascent.* The observation of the WCB cloud band in the northern part of the WCB outflow at 16:00 UTC emphasizes its predominantly stratiform character (Fig. 2b), which coincides with a region of comparatively slowly ascending WCB trajectories with $\Delta p_{2h}$ in the range from $-25$ to $-50$ hPa (Figs. 2a, orange line and 2b,





grey shading). The large-scale cloud band, extending to about 9 km height, is characterized by a relatively homogeneous radar reflectivity, a closed upper-level cloud top and a well-defined bright band at 2 km height. This extended cloud band is associated with WCB ascent in the mid- to upper troposphere between 4 km and 9 km height. The mostly stratiform character of this cloud band is confirmed by the IFS dataset, which produces hardly any convective precipitation along the flight track (Fig. 2b, bottom panel), but moderate large-scale precipitation of up to 2.5 mm h$^{-1}$ (not shown).

*Convective plumes associated with WCB ascent.* The WCB cloud band further south at 48°N near the cyclone center (Fig. 2c) differs substantially from the stratiform cloud structure observed in the northern part of the WCB cloud band (Fig. 2b). The radar reflectivity $Z$ in this region is very heterogeneous with spatially confined narrow convective plumes that extend from the boundary layer to 6-8 km height and locally exceed the cloud top height in their direct surroundings, which accentuates the convective character (Fig. 2c). Due to these vertically extended narrow convective towers, the cloud top is heterogeneous and

cloud top height locally varies between 5-8 km. The radar reflectivity $Z$ within the convective plumes amounts to approximately 25 dB$Z$, with values ranging from 20-25 dB$Z$ even in the upper troposphere several kilometers above the melting layer. Hence, although $Z$ does not exceed 25 dB$Z$, the embedded convective updrafts can be clearly identified as individual narrow plumes, which directly coincide with WCB air masses between 1 km and 6 km height. This qualitatively confirms results from previous studies (Crespo and Posselt, 2016; Oertel et al., 2019; Blanchard et al., 2020), although absolute values of radar reflectivity are

difficult to compare since in strongly precipitating regions attenuation effects through Mie scattering have to be considered at shorter wavelengths (e.g., Ewald et al., 2019). This affects systems like the 94 GHz CloudSat radar (analysed by Crespo and Posselt, 2016) and the 95 GHz RASTA radar (analysed by Blanchard et al., 2020) more strongly than the here used MIRA-36, which uses a slightly larger wavelength.

Near the observed convective plumes, COSMO WCB trajectories frequently perform a convective ascent with $\Delta p_{2h} <$

$-400$ hPa (Figs. 2a, yellow line and 2c, grey shading). These phases of convective WCB ascent aggregate northeast of the cyclone center and are directly embedded in a region of slantwise WCB ascent with $\Delta p_{2h}$ values between $-50$ and $-150$ hPa. Near the flight track, additional regions of convective WCB ascent with $\Delta p_{2h} < -320$ hPa occur near the warm and cold fronts. The convective character of the WCB cloud band is corroborated by the production of substantial convective precipitation by the IFS parameterization scheme (Fig. 2c, bottom panel). The comparison between the observations from both flight legs em-

phasizes the diverse cloud structure associated with the WCB that can range from a homogeneous, large-scale, closed-top cloud band (Fig. 2b) to a heterogeneously structured cloud with embedded convective plumes and a broken cloud top (Fig. 2c).

*Weaker convection embedded in extended WCB cloud band.* The flight on 10 October sampled the WCB ascent and outflow region twice during two subsequent west-east traverses (Fig. 3a, yellow line) and provides another example of embedded convection in the WCB ascent region (Fig. 3b). Near the flight track, WCB air parcels ascend with $\Delta p_{2h}$ values between

$-250$ and $-320$ hPa with few WCB air parcels with $\Delta p_{2h}$ below $-320$ hPa (Figs. 3a and 3b, grey shading). These ascent rates are faster than a purely slantwise ascending WCB trajectory ($\Delta p_{2h} \approx -50$ hPa, Fig. 2b), but the convective WCB ascent phases are slightly slower and less frequent than the extended region of convective WCB ascent near the cyclone center on 09 October (Fig. 2c). In the radar observations the convective plumes at around 14:30 UTC and 15:15 UTC, which also approximately coincide with parameterized convective precipitation and small $\Delta p_{2h}$ values, are less well-defined and stand





out less (Fig. 3b) compared to the extended region with embedded convection near the cyclone center (Fig. 2c). Moreover, the regions of faster WCB ascent, characterized by narrow locally confined regions of enhanced radar reflectivity, are directly embedded in the larger-scale vertically elevated cloud band with stratiform characteristics, such as regions of homogeneous radar reflectivity and a bright band which is visible in most regions. Nevertheless, the large-scale WCB cloud band is more heterogeneous than the stratiform cloud band in the northern part of the WCB (Fig. 2b) and includes confined plumes of
locally increased radar reflectivity that extend up to 6-8 km height, which indicates the occurrence of locally faster WCB ascent associated with embedded convective activity.

The comparison of the radar reflectivity cross-sections from the first (Fig. 2b,c) and the second flight (Fig. 3b) illustrates the large spatial and temporal heterogeneity of the WCB cloud band. On the one hand, the extended and dense WCB cloud band
has a stratiform character associated with primarily slantwise WCB ascent. On the other hand, it includes embedded convection in the WCB ascent region very close to the cyclone center, at a distance of about 300 km from the stratiform region. Moreover, the various degrees of heterogeneity of the radar reflectivity structure – which reflect the strength of embedded ascent phases – illustrate the almost continuous transformation from the slantwise to the convective WCB ascent regime (see Oertel et al., 2019). Hence, the radar observations emphasize the diverse representation and manifold character of the WCB cloud band
including its embedded convection, thereby corroborating its heterogeneity.

The comparison between the airborne radar observations and the online trajectories from the convection-permitting simulation shows that qualitatively the COSMO online trajectories cannot only successfully identify regions of embedded convection in the WCB ascent region, but can also differentiate between different degrees of the strength of embedded convection. In summary, regions with stronger convective updrafts include well-defined individual plumes of increased $Z$ and are characterized by large
spatial heterogeneity of $Z$. These regions also coincide with abundant convective WCB ascent in the COSMO simulation with $\Delta p_{2h} < -400$ hPa. Weaker embedded convection, however, stands out less against the large-scale stratiform cloud band, but can nevertheless be identified by enhanced heterogeneity and less coherent, vertically extended regions of increased $Z$. In the COSMO simulations, this weaker embedded convection can be identified through intermediate $\Delta p_{2h}$ values, which are substantially larger than for a pure slantwise ascent, but still slower than the stronger convective updrafts observed near the
cyclone center. In the following, we extend the analysis of embedded convection beyond the region where HALO observed the WCB cloud band, i.e., we also include very intense convection in the early phase of the cyclone with $\Delta p_{2h} < -600$ hPa (Fig. 1j).



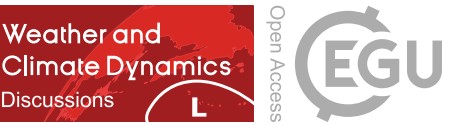

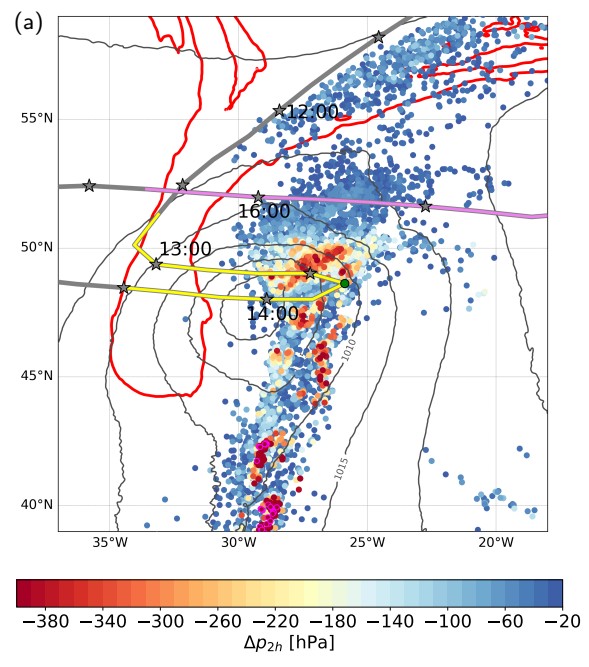

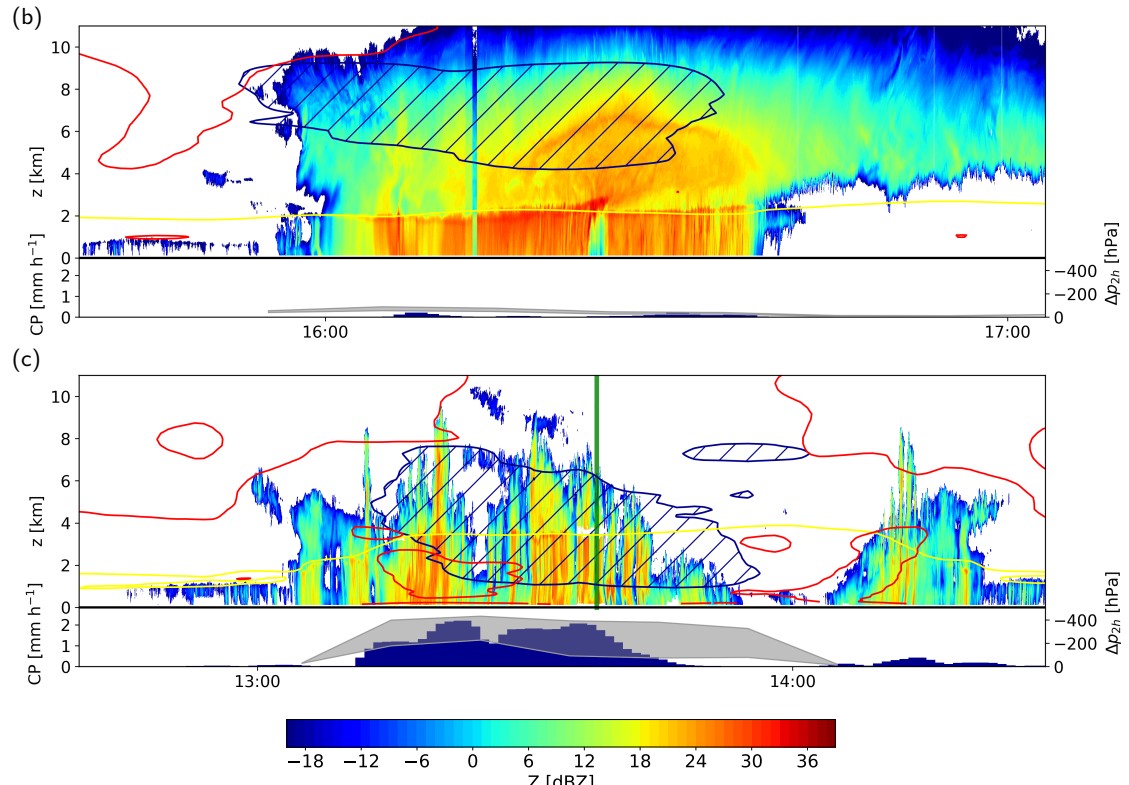





**Figure 2. (a)** HALO flight track (grey) on 09 October 2016 including 2-h pressure changes ($\Delta p_{2h}$, colored points) of COSMO WCB trajectories centered at 13:30 UTC, SLP (grey contours, every 5 hPa) and 2 PVU at 320 K (red). COSMO WCB air parcels ascending more than 600 hPa in 2 h are shown by magenta encircled dots. The grey asterisks indicate the position of the aircraft every 30 minutes. Yellow and violet parts of the flight track denote the cross-sections shown in **(b,c)** from 15:40 to 17:00 UTC and from 12:40 to 14:30 UTC, respectively. **(b,c)** Radar reflectivity factor $Z$ (in dB$Z$) along **(b)** the violet segment and **(c)** the yellow segment in **(a)** including WCB mask from ECMWF WCB trajectories (dark blue contour and hatching), the $0\,^{\circ}$C isotherm (yellow) and the 2 PVU contour (red). The bottom panels show ECMWF convective precipitation (CP, in mm h$^{-1}$) along the flight track (blue bars), and range of 25% fastest 2-h ascent phases of COSMO WCB air parcels (i.e., 25% smallest $\Delta p_{2h}$ values) shown in **(a)** within a radius of 75 km around the aircraft position (in hPa, grey shading). The green line in **(c)** at 13:40 UTC marks the aircraft turning point east of the cyclone center and is marked as green dot in **(a)**.

L



**Figure 3. (a)** As Fig. 2a but on 10 October 2016 with COSMO online trajectories centered at 15:00 UTC. The cross-section shown in **(b)** is marked in yellow (14:10-15:40 UTC). **(b)** As Fig. 2b but on 10 October from 14:10-15:40 UTC. Also shown is the aircraft height (black). The green line at 14:50 UTC marks the eastern aircraft turning point and is marked as green dot in **(a)**.



## 5 Characteristics of intense vs. moderate WCB-embedded convection

Radar observations and online WCB trajectories showed the frequent occurrence of convection with differing ascent speed embedded in the WCB of cyclone Sanchez. In the following, we systematically compare characteristics of intense vs. moderate

WCB-embedded convection in the COSMO simulation. In addition to $\Delta p_{2h}$, we now use two alternative measures to describe the WCB trajectory ascent speed in the entire cyclone, which allow for a more detailed distinction of the intensity of convection and are particularly useful for comparison of ascent speed with other studies (e.g., Carbone, 1982; Flaounas et al., 2016; Oertel et al., 2019; Oertel, 2019; Blanchard et al., 2020). These are the minimum time required for an ascent of 400 and 600 hPa ($\tau_{400}$ and $\tau_{600}$), respectively, and ascent speed ($\omega$, in Pa s$^{-1}$ and w, in m s$^{-1}$) averaged over these 400 and 600 hPa segments.

$\tau_{400}$ and $\tau_{600}$ values demonstrate the large variability of ascent rates for the WCB and its embedded convection ranging from below 1 h to more than 24 h (Fig. 4). The frequency distributions for $\tau_{400}$ (Fig. 4a) and $\tau_{600}$ (Fig. 4b) are bimodal with a first local maximum at short durations of about 30 min and 1 h, respectively, and a second maximum at 2.5 h and 6 h, respectively. Interestingly, the second maximum of $\tau_{400}$ of 2.5 h corresponds to a previously used threshold for embedded convection (Rasp et al., 2016; Oertel et al., 2019) separating the slantwise from the convective WCB ascent regime. The 10% lowest $\tau_{400}$ and

$\tau_{600}$ values for all WCB trajectories are below 20 minutes and 1 h, respectively. This left tail of the distribution shows that embedded convective WCB ascent with averaged $\omega$ of approximately 17-33 Pa s$^{-1}$ (corresponding to $\tau_{400}$ = 20 min and $\tau_{600}$ = 1 h) occur quite often in this cyclone. In terms of w, the ascent velocity of embedded convection amounts to approximately 3-5 m s$^{-1}$. This is about two orders of magnitude faster than a hypothetical purely slantwise ascending WCB trajectory with w = 0.06 m s$^{-1}$ ($\omega$ = 0.3 Pa s$^{-1}$; assuming a continuous 48-h ascent of 600 hPa or approximately 10 km), and also exceeds the

previously reported ascent velocities of WCB-embedded convection (ranging between 1-17 Pa s$^{-1}$, e.g., Flaounas et al., 2016; Rasp et al., 2016; Oertel et al., 2019, 2020; Blanchard et al., 2020). The ascent velocities indicate that faster and even more intense convection – with similar ascent velocities as deep convective ascent at squall lines (Rasp et al., 2016) – can also be embedded in WCBs.

### 5.1 Lagrangian composite analysis

In the following, we perform a 3D Lagrangian composite analysis following the method of Oertel et al. (2020) to compare the characteristics and signatures of relatively moderate and intense convection embedded in the WCB of cyclone Sanchez (Fig. 5). During the evolution of the surface cyclone, both the upper-level flow structure changes and the strength of convective WCB ascent differs (see Sect. 3.3). In particular, embedded convection in the early phase (Fig. 1j) is much more intense than in the later phase (Fig. 1k,l). However, the compositing technique requires a coherent trajectory ascent from the lower to the upper

troposphere and a similar synoptic-scale flow situation to not smear out the signals. Hence, we compare two sub-categories of convective WCB trajectories that ascend coherently in two sub-regions of the WCB: (i) very rapidly ascending, intense convective WCB trajectories, and (ii) moderately ascending (but still) convective WCB trajectories (Fig. 5). In the following, the selection criteria for both convective WCB sub-categories are described in more detail.



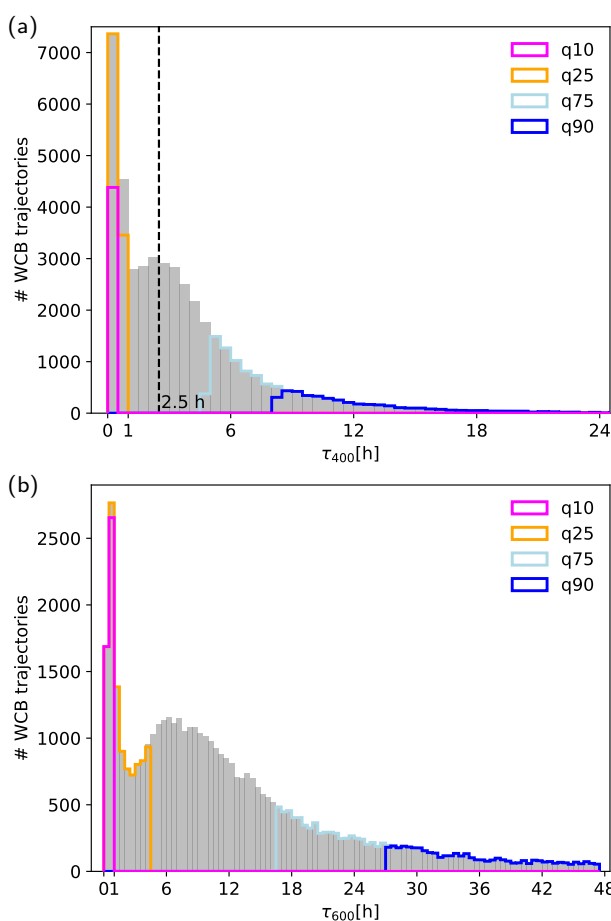

**Figure 4.** Histogram of the duration of the fastest **(a)** 400-hPa ascent phases ($\tau_{400}$) and **(b)** 600-hPa ascent phases ($\tau_{600}$) embedded in all WCB trajectories of cyclone Sanchez. The $10^{th}$, $25^{th}$, $75^{th}$ and $90^{th}$ percentiles are colored in magenta, orange, light blue and blue, respectively. The dashed line at $\tau_{400} = 2.5\,h$ in **(a)** indicates a previously used threshold for WCB-embedded convection (Rasp et al., 2016; Oertel et al., 2019). Note the different axes dimensions in **(a)** and **(b)**.

## 5.2 Trajectory selection and ascent region

*(i) Intense convective WCB trajectories.* The selected intense convective WCB trajectories ascend through the entire troposphere in less than 1 h (Fig. 5). On average, these trajectories ascend from 1 km to 9 km height in only 30 minutes and reach their final outflow level at 11 km height within 1 h. Their $\tau_{400}$ and $\tau_{600}$ values amount to 20 minutes and 1 h, respectively (corresponding to the $10^{th}$ percentile of all $\tau_{400}$ and $\tau_{600}$ values for cyclone Sanchez). This translates to averaged 400 and 600 hPa ascent velocities of 33 Pa s$^{-1}$ and 17 Pa s$^{-1}$, respectively, i.e., this sub-category represents intense deep convection with clearly

faster ascent velocities compared to WCB-embedded convection of cyclone Vladiana, which occurred two weeks earlier in September 2016 (Oertel et al., 2020). In particular, the sub-category also includes convective WCB trajectories with the short-

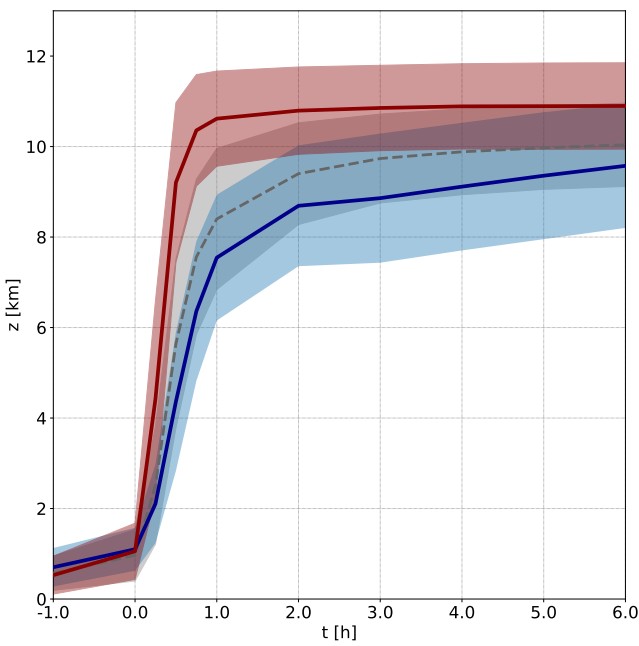

**Figure 5.** Mean ascent (solid lines; shading shows mean $\pm$ standard deviation) of selected moderate convective (blue) and intense convective (red) WCB trajectories of cyclone Sanchez that are used for the composite analysis. For comparison, the ascent of embedded convective WCB trajectories of cyclone Vladiana investigated by Oertel et al. (2020) is also shown (grey).

est $\tau_{600}$ of only 30 min, i.e., the WCB trajectories traverse the entire 600 hPa layer from the boundary layer into the upper troposphere with an average ascent rate of 33 Pa s$^{-1}$ (6 m s$^{-1}$). Approximately 2000 trajectories fall in the category of intense convective WCB trajectories. The intense convective WCB trajectories ascend from 07-09 October predominantly between 35-

40°N (on average at 38°$\pm$2.0°; Fig. 6a, black outlined circles), as during this early stage of the cyclone, the WCB-embedded intense convection occurs further south and also closer to the cyclone center (Fig. 1j).

*(ii) Moderate convective WCB trajectories.* The trajectories in the moderate convective WCB sub-category are spatially and temporally separated from the intense convective WCB trajectories (Fig. 6a). They ascend north of 40°N (on average at 43°$\pm$1.8°) between 12 UTC 09 October and 12 UTC 11 October during the PV cutoff phase (Fig. 6a, white outlined

triangles). During this time period convective WCB ascent is overall slower than in the early phase of the cyclone (Fig. 1j-l). Hence, the moderate convective WCB trajectories are characterized by a slower ascent (Fig. 5), whereby the trajectories perform an initial convective 400 hPa deep ascent from 1 km to approximately 8 km height within less than 1 h (corresponding to the 25$^{th}$ percentile of all $\tau_{400}$ values for cyclone Sanchez) with an average ascent velocity of 11 Pa s$^{-1}$ (Fig. 5). In the next 2-3 h, their cross-isentropic ascent velocity is reduced and they finish their full 600 hPa ascent at 10 km height in less

than 9 h (corresponding to the median of all $\tau_{600}$ values for cyclone Sanchez), which amounts to an averaged ascent velocity of 2 Pa s$^{-1}$. Compared to WCB-embedded convection of cyclone Vladiana (Oertel et al., 2020), the ascent of the moderate





convective WCB trajectories is slightly slower (Fig. 5). In total, approximately 1000 WCB trajectories are selected for this sub-category.

For the composite analysis the selected WCB trajectories from both sub-categories are centered relative to the start of their fastest embedded 400 hPa ascent phase in the lower troposphere. Subsequently, three types of composites are produced for both convective WCB sub-categories: composites of (i) horizontal and (ii) vertical cross-sections centered at the trajectories' geographical position, and (iii) composites of vertical profiles along the trajectories, i.e. time-height sections along the flow. A horizontal cross-section composite of low-level $\theta_e$ illustrates the ascent region of the moderate convective WCB trajectories

ahead of the cold front southeast of the cyclone center (Fig. 6b). The edge of the upper-level trough (PV cutoff) and the associated weak upper-level jet are located in approximately 200 km distance northwest of the initial trajectory ascent region (Fig. 6b). In contrast, the ascent of the intense convective WCB trajectories is located closer to the cyclone center compared to the moderate convective WCB trajectories (Fig. 6c). They ascend ahead of the cold front only about 150 km south of the cyclone center. When these intense convective WCB trajectories start their ascent south of 40°N before 10 October, the upper-

level trough and accompanying polar jet are located much further north at approximately 50°N (Fig. 1j).

## 5.3 Cloud and precipitation structure

The cloud and precipitation structure associated with the selected WCB trajectories corroborates their convective character (Fig. 7a-d). The initial fast ascent forms a locally dense cloud with enhanced rain and snow production along the ascent (Fig. 7a,b), whereby the hydrometeor formation for the intense convective WCB trajectories is substantially larger than for the

moderate convective WCB trajectories. In the mid-troposphere, between 4-7 km height, the convective ascent allows for the formation of large amounts of graupel. Due to the stronger updrafts of the intense convective WCB trajectories, the graupel production even extends to 11 km height (Fig. 7b). The maximum graupel water content (GWC) in the mid-troposphere at approximately 320 K exceeds 2.8 g kg$^{-1}$ and is more than twice the maximum GWC produced by the moderate convective WCB trajectories with GWC of 1.0 g kg$^{-1}$. Both sub-categories of embedded convection form a localized vertically-extended

dense cloud with elevated cloud top. Thereby, the ice water content peaks directly above the mean trajectory position, forming a dense cirrus shield, which is particularly dense for the intense convective WCB sub-category with IWC>0.1 g kg$^{-1}$ compared to IWC≃0.05 g kg$^{-1}$ for the moderate convective sub-category. Once the WCB trajectories are located in the upper troposphere they remain inside a cirrus cloud for at least the next 6 h. In agreement with the large hydrometeor production, the surface precipitation peaks during the strongest WCB ascent phase. The moderate convective WCB trajectories reach a maximum

of 3.9 mm of 15 minute accumulated surface precipitation (Fig. 7c), while for the intense convective WCB trajectories, the maximum value exceeds 6 mm (Fig. 7d).



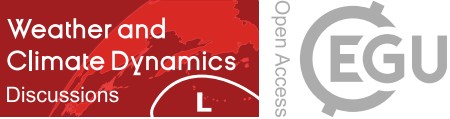

**Figure 6. (a)** Location of moderate convective (white outlined triangles) and intense convective (black outlined circles) WCB trajectories at the start of the fastest 400-hPa ascent phase. Colors indicate the according time of the WCB air parcel position (from 07 to 10 October 2016). Also shown is the according evolution of SLP (lines are colored according to the selected times every 12 h from 00 UTC 08 October to 12 UTC 10 October). **(b,c)** Horizontal cross-section composites of equivalent potential temperature ($\theta_e$, colors, in K), specific humidity (blue contours, every $1\,\mathrm{g\,kg^{-1}}$) and wind vectors at 900 hPa (arrows) for **(b)** moderate convective and **(c)** intense convective WCB trajectories at the start of the fastest 400 hPa-ascent. In **(b)** the upper-level jet (green colors, at 30 and $33\,\mathrm{m\,s^{-1}}$) and the 2 PVU contour at 320 K (white-red line) are also shown. The axes' dimensions denote the distance from the WCB air parcel locations marked as '$\times$' (in km).



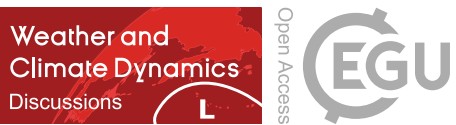

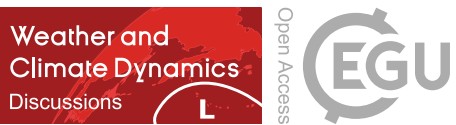





**Figure 7.** Composites of vertical profiles following the motion of the trajectories (black line shows the mean ascent of all WCB trajectories) for **(a,c,e,g)** moderate and **(b,d,f,h)** intense convective WCB trajectories: **(a,b)** hydrometeors [sum of rain and snow water content (RWC + SWC, colors, in $\mathrm{g\,kg^{-1}}$), ice water content (IWC, turquoise contours, every $0.02\,\mathrm{g\,kg^{-1}}$), graupel water content (GWC, magenta contours, every $0.4\,\mathrm{g\,kg^{-1}}$], $0\,°\mathrm{C}$-isotherm (yellow line), $320\,\mathrm{K}$ isentrope (blue line) and the $2\,\mathrm{PVU}$ tropopause (red line); **(c,d)** $15$ minute accumulated surface precipitation along the ascent (blue, in mm); **(e,f)** moist stability ($\mathrm{d}\theta_e/\mathrm{d}z$, colors, in $\mathrm{K\,km^{-1}}$), the $320\,\mathrm{K}$ isentrope (blue line), and relative humidity (RH, grey contours, in %; 97% and 99% RH contours are highlighted in green and lime); **(c,d)** CAPE along the ascent (orange, in $\mathrm{kJ\,kg^{-1}}$).

The formation of a locally dense cloud with enhanced hydrometeor content, the presence of graupel in the mid-troposphere, and locally increased surface precipitation underlines the convective character of the selected WCB trajectories for both sub-categories. These characteristics are comparable with the previous case study of WCB-embedded convection in cyclone Vladiana (Oertel et al., 2020), where the vertically extended dense cloud was accompanied by a local precipitation peak of 4.5 mm of 15 minute accumulated surface precipitation and substantial graupel production in the embedded convective updraft. The differences between both WCB sub-categories emphasizes – that similarly to the observed differences between both convective radar reflectivity cross-sections (cf. Figs. 2c and 3b) – the intensity of WCB-embedded convection can vary substantially within one WCB. This varying strength has a direct impact on the local cloud structure, the hydrometeor content, and the surface precipitation (Table 1).

### 5.4 Thermodynamic properties

Both sub-categories of convective WCB trajectories originate from an almost saturated (Fig. 7e,f), warm, and moist region in the lower troposphere (Fig. 6b,c). The inflow region is characterized by a deep layer of strong potential instability with vertical $\theta_e$ gradients between $-2$ and $-4\,\mathrm{K\,km^{-1}}$ prior to the start of the convective ascent (Fig. 7e,f). The higher initial $\theta_e$ of $337\,\mathrm{K}$ and the presence of stronger potential instability for the intense convective WCB trajectories enable their larger cross-isentropic ascent from $299\,\mathrm{K}$ to $336\,\mathrm{K}$ ($11\,\mathrm{km}$) in only $1\,\mathrm{h}$ (in agreement with their initial $\theta_e$) and their faster ascent velocity compared to the moderate convective WCB trajectories with an initial $\theta_e$ of $332\,\mathrm{K}$ and a corresponding reduced outflow level at on average $330\,\mathrm{K}$ ($9\,\mathrm{km}$) after $6\,\mathrm{h}$. Thus, the intense convective WCB trajectories do not only reach their final height substantially faster, but their isentropic outflow level is also approximately $6\,\mathrm{K}$ higher than the outflow level of the moderate convective WCB trajectories. In contrast to the intense convective WCB trajectories, which are associated with the presence of high convective available potential energy (CAPE) values of on average $1000\,\mathrm{J\,kg^{-1}}$ prior to their ascent (Fig. 7h), the moderate convective WCB trajectories ascend in a region characterized by very low values of CAPE (Fig. 7g). The absence of high CAPE values during convective WCB ascent agrees with the previous WCB case study in cyclone Vladiana, where the WCB inflow and ascent region was characterized by comparatively low values of CAPE but substantial large-scale forcing for ascent (Oertel et al., 2019; Oertel, 2019, Fig. 4.3c).



**Table 1.** Characteristics of WCB-embedded convection for (i) moderate and (ii) intense convection in cyclone Sanchez (this study), and (iii) convection in cyclone Vladiana (Oertel et al., 2020). Shown are averaged trajectory ascent in the first hour after start of the ascent, maximum 15 minute accumulated surface precipitation, maximum graupel water content (GWC), convective available energy (CAPE) prior to the ascent, mean magnitude of negative PV pole at 320 K and its position relative to the convective updraft, and magnitude and direction of vertical wind shear in the middle- to upper troposphere.

|  | moderate convection in Sanchez | intense convection in Sanchez | convection in Vladiana |
|---|---|---|---|
| ascent | 400 hPa in 1 h | 600 hPa in 1 h | 550 hPa in 1 h |
| precipitation | 3.9 mm | >6 mm | 4.5 mm |
| GWC | 1.0 g kg$^{-1}$ | 2.8 g kg$^{-1}$ | 1.8 g kg$^{-1}$ |
| CAPE | 90 J kg$^{-1}$ | 1000 J kg$^{-1}$ | 290 J kg$^{-1}$ |
| negative PV pole | −0.3 PVU | −0.2 PVU | −1.5 PVU |
| position | northeast | east | northwest |
| wind shear | weak | weak | strong |
| direction | southeast | south | northeast |

## 5.5 PV structure

To find the characteristic upper-level PV dipole structure of WCB-embedded convection (e.g., Oertel et al., 2020), horizontal
composites centered around the convective ascent are considered (Fig. 8a,b). These composites are evaluated at the 320 K isentrope which corresponds approximately to the level of maximum diabatic heating associated with the maximum hydrometeor content formation (which is dominated by graupel production, Fig. 7a,b). Both convective WCB sub-categories form a similarly weak and small-scale horizontal PV dipole, whereby one pole reaches negative PV values. The negative PV pole with a diameter of 5-10 km is located to the left of the vertical wind shear vector (a more detailed discussion follows in Sect. 5.6),
i.e., northeast of the ascent region for the moderate and east of the ascent region for the intense convective WCB trajectories, respectively (Fig. 8a,b).

A vertical cross-section through the PV dipole for the moderate convective WCB trajectories (Fig. 8d) reveals a mid- to upper-level PV dipole that is tilted away from the vertical and where the negative PV pole with an average magnitude of −0.3 PVU extends from 5 km to 9 km height. The intense convective WCB trajectories form a mid- to upper-level PV dipole between 5-
8 km height (where the negative PV pole reaches −0.2 PVU on average), that is located above a positive low-level PV anomaly in the lowermost 4 km (Fig. 8e). Compared to the moderate convective WCB sub-category, the tilted quasi-vertical PV dipole tower is slightly more pronounced, and especially the low-level positive PV anomaly is vertically more extended. [2] The PV

---

[2]The vertically extended positive low-level to mid-level PV monopole of magnitude 3.7 PV formed by the intense convective WCB trajectories (Fig. 8d) is located near the cyclone center (Fig. 6c), and, thus, generates and contributes to the positive PV anomaly in the cyclone center potentially favoring the cyclone's maintenance and subsequent intensification (cf. Binder et al., 2016).





dipole is covered by a pronounced region of low-PV air that even includes a region of negative PV at 335 K (10 km height). Hence, the low-PV pole of the dipole is weaker and the negative PV is scattered in multiple elements.

Although a PV dipole forms for both sub-categories of embedded convection, the detailed PV dipole structure differs substantially from the much more pronounced PV dipole tower in cyclone Vladiana (Fig. 8c,f, and Oertel et al., 2020). To understand the differences, the main results for cyclone Vladiana by Oertel et al. (2020) are summarized in the following: Embedded convection in the WCB of cyclone Vladiana, with an average ascent of 600 hPa in 3 h (Fig. 5), consistently occurred in a narrow band ahead of the cold front and the upper-level trough with a strong upper-level jet of approximately 50 m s$^{-1}$. Although em-

bedded convection was weaker than the intense embedded convection in this case (Fig. 5), the convective WCB ascent formed a coherent vertically-extended PV dipole tower between 3-9 km height with a stronger maximum amplitude of the PV dipole of $-1.5$ PVU and 3 PVU across both poles at around 320 K (approximate level of maximum hydrometeor content). Moreover, the upper-level PV dipole had a larger diameter of approximately 30 km and was oriented from the northwest to the southeast, with the negative PV pole to the left of the northeastward pointing vertical wind shear vector, i.e., in vicinity to the upper-level

waveguide (Fig. 8c,f).

Hence, compared to the composite PV structure of WCB-embedded convection in cyclone Vladiana (Fig. 8c,f), two major differences arise (Fig. 8): (i) in cyclone Sanchez, the upper-tropospheric PV dipole has a reduced horizontal and vertical extent and the magnitude of the PV dipole is weaker (despite faster convective ascent for the intense convective WCB trajectories in cyclone Sanchez compared to the convective WCB ascent in cyclone Vladiana); and (ii) at 320 K, the PV dipole is oriented

in the opposite direction, i.e., the negative PV pole is located northeast/east of the convective updraft, i.e., away from the upper-level waveguide (vice versa for the PV tower in cyclone Vladiana, where the negative PV pole was located close to the upper-level waveguide; see also Harvey et al., 2020).

The common structure in all three composites is the location of the negative PV pole to the left of the vertical wind shear vector

(Fig. 8a-c). In agreement with theoretical considerations (Harvey et al., 2020; Oertel et al., 2020), the horizontal PV dipole forms if the horizontal vorticity vector $\boldsymbol{\omega_h}$ and the horizontal diabatic heating gradients ($\nabla_h \dot{\theta}$) are aligned, such that PV is produced where $\nabla_h \dot{\theta} \parallel \boldsymbol{\omega_h}$ and is destroyed where $\nabla_h \dot{\theta} \parallel -\boldsymbol{\omega_h}$ (see Eq. 1). Thus, the resulting PV structure is substantially governed by $\boldsymbol{\omega_h}$, which is rotated 90° anticlockwise relative to the vertical wind shear vector $\mathrm{d}\mathbf{v}/\mathrm{d}z$ and scales with $|\mathrm{d}\mathbf{v}/\mathrm{d}z|$. [3] The differences in the PV structure of WCB-embedded convection between both types of convective WCB ascent in cyclone

Sanchez, and the PV structure of WCB-embedded convection in cyclone Vladiana (Oertel et al., 2020), are directly related to the differences in the vertical wind shear profiles and the relative alignment of embedded convection and the upper-level jet. This is investigated in more detail in the next subsection.

---

[3]According to Eq. 3 and neglecting the horizontal gradients of $\boldsymbol{\omega}$, we get $\boldsymbol{\omega_h} = (-\frac{dv}{dz})\mathbf{i} + (\frac{du}{dz})\mathbf{j}$, and $|\boldsymbol{\omega_h}| \approx \sqrt{(\frac{dv}{dz})^2 + (\frac{du}{dz})^2}$.

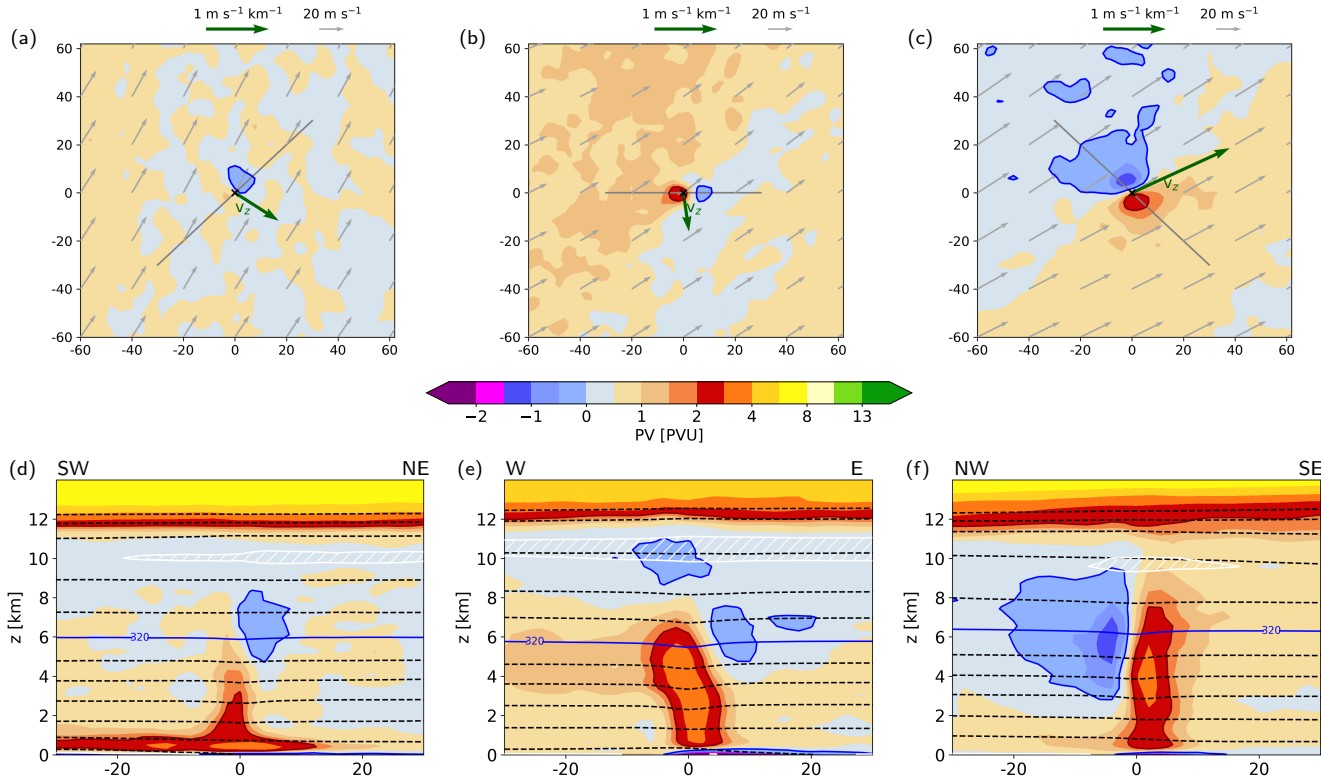

**Figure 8. (a,b,c)** Horizontal cross-section composites of PV (colors, in PVU) and wind speed (grey arrows, in m s$^{-1}$) at 320 K for **(a)** moderate and **(b)** intense convective WCB trajectories in cyclone Sanchez **(a)** 30 minutes after the start of the fastest 400-hPa ascent and **(b)** 15 minutes after the start of the fastest 400-hPa ascent. For comparison, **(c)** shows convective WCB trajectory ascent in cyclone Vladiana (Oertel et al., 2020). The green arrow shows the vertical wind shear vector ($\boldsymbol{v_z}$) between 5 and 7 km height at the start of the WCB trajectory ascent. The axes' dimensions denote the distance from the WCB air parcel locations marked as '×' (in km). **(d,e,f)** Vertical cross-section composites along the lines shown in **(a,b,c)**, respectively, of PV (colors, in PVU), isentropes (dashed lines, every 5 K), the 320 K isentrope (blue line) and low static stability layers (d$\theta$/dz $\leq 2$ K km$^{-1}$, white contour and hatching). The convection-permitting simulation and Lagrangian composite analysis for cyclone Vladiana was performed in a similar way as in this study. For details see Oertel et al. (2020).

## 5.6  Vertical wind shear profiles in the convective ascent regions

Both embedded convective regions in the WCB of cyclone Sanchez occur in a region characterized by weak vertical wind shear

(Fig. 9). For the moderate convective WCB trajectories, wind speed between 1-6 km height is almost constant (Fig. 9a), and only above 7 km wind speed increases resulting in a mean vertical wind shear of 1-2 m s$^{-1}$ km$^{-1}$ that is directed towards the northeast (Fig. 9a,b). At 6 km, the approximate level of the diabatic heating maximum (Fig. 7a), the vertical wind shear has a magnitude of approximately 1 m s$^{-1}$ km$^{-1}$, and the wind shear vector points to the southeast. Hence, the horizontal vorticity vector $\boldsymbol{\omega_h}$ rotates with height accordingly, which in turn modulates the PV modification according to $\omega_h \cdot \nabla_h \dot{\theta}$, and results in

a northeast-southwest oriented PV dipole at 6 km height (Fig 8a,d).





For the intense convective WCB trajectories, with ascent far south of the upper-level jet, the wind speed hardly varies with height throughout almost the entire tropospheric column from 2-11 km height (Fig. 9a), resulting in very weak vertical wind shear (Fig. 9b). Indeed, between 4-8 km the wind speed even slightly decreases with height, such that the vertical wind shear vector points towards the south. Accordingly, the west-east oriented PV dipole forms at 6 km height with negative PV to the

east of the convective updraft, i.e., left of the vertical wind shear vector (Fig. 8b,e). Hence, despite stronger diabatic heating and associated stronger horizontal heating gradients $\nabla_h \dot{\theta}$ for the intense convective WCB sub-category, the PV dipole amplitude and size is not larger compared to the moderate convective WCB sub-category due to the weak vertical wind shear in the middle to upper troposphere.

  These differing vertical wind profiles result from the synoptic situation in which the convective WCB ascent is embedded.

The intense convective WCB trajectories ascend in a region characterized by the absence of a strong upper-level jet (Figs. 1b and 6c). The upper-level wind speed hardly exceeds 20-25 m s$^{-1}$, which is similar to the wind speed of the low-level jet (Fig. 9a). The moderate convective WCB trajectories ascend further north during the PV cutoff phase (Figs. 1a and 6b) and are located southeast of an upper-level jet of 30-35 m s$^{-1}$ at 11 km height. Hence, at least in the upper troposphere, the wind speed increases leading to comparatively stronger vertical wind shear (Fig. 9a). For comparison, the northwest-southeast oriented PV

dipole of embedded convection in cyclone Vladiana (Oertel et al., 2020) with a magnitude of −1.5 PVU and 3 PVU across both poles is located in a region characterized by coherent unidirectional wind shear of magnitude 3 m s$^{-1}$ km$^{-1}$ in a deep layer below the upper-level jet pointing to the northeast (Fig. 9a,b). The alignment of embedded convection and the vertical wind shear in the WCB of cyclone Vladiana gives rise to the pronounced PV dipole structure with negative PV at the jet-facing side. We conclude that the PV dipole structure associated with convective WCB ascent is strongly modulated by the large-scale flow,

as expected from theoretical considerations. Despite the more intense convective ascent and the associated increased localised diabatic heating, the resulting PV anomalies are not stronger in cyclone Sanchez (cf. Figs. 8a,b). In contrast, compared to cyclone Vladiana (Oertel et al., 2020) the associated PV anomalies of the intense convective WCB ascent in this study are weaker and the mid- to upper-level PV dipole extends over a smaller spatial scale – despite faster convective ascent. This implies that the upper-level PV modification by WCB-embedded convection and its relevance for the large-scale flow evolution can

differ substantially from case to case. More specifically, the comparison of the PV structures associated with cyclone Sanchez (this study) and cyclone Vladiana (Oertel et al., 2020) emphasizes that the intensity of embedded convection alone is not a reliable measure for the degree of PV modification (Table 1). Instead, the alignment of WCB-embedded convection and the upper-level jet determine the potential to produce coherent upper-level negative PV features near the jet, which were found to influence the upper-level jet and downstream flow evolution in the strong-jet situation of cyclone Vladiana (Oertel et al., 2020).



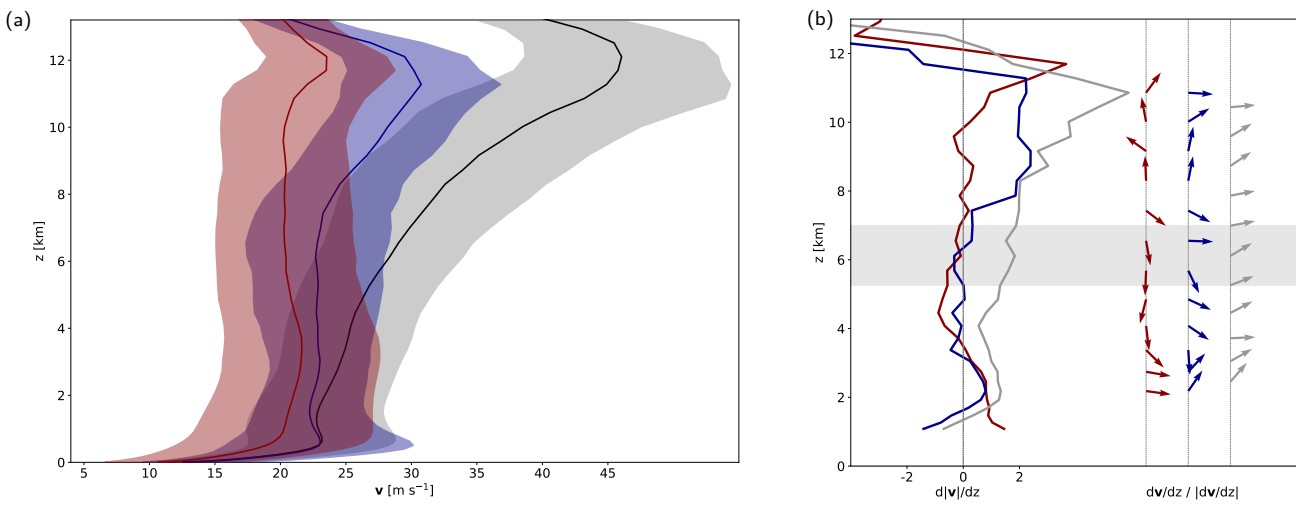

**Figure 9. (a)** Mean vertical profile of the horizontal wind speed ($\mathbf{v}$, solid lines, in $\mathrm{m\,s^{-1}}$) including the standard deviation (shading) in the region of convective WCB ascent for the moderate convective (blue) and intense convective (red) WCB ascent in cyclone Sanchez and for convective WCB ascent in cyclone Vladiana (grey Oertel et al., 2020); **(b)** magnitude of vertical wind shear ($\mathrm{d}|\mathbf{v}|/\mathrm{d}z$, in $\mathrm{m\,s^{-1}\,km^{-1}}$) and direction of the vertical shear vectors of the horizontal wind standardized by their magnitude ($\mathrm{d}\mathbf{v}/\mathrm{d}z/|\mathrm{d}\mathbf{v}/\mathrm{d}z|$, colored arrows) for selected altitudes. Colors as in **(a)**. The 5-7 km layer (approximate level of 320 K isentrope) is outlined in grey for comparison with Fig. 8.



## 6 Conclusions

This study corroborates the presence of enhanced ascent phases and embedded convection in the large-scale WCB cloud band based on airborne radar observations and online trajectories from a convection-permitting simulation.

The first part of this study focuses on the identification of WCB-embedded convection with differing intensity associated with cyclone Sanchez. Overall, this study confirms the frequent occurrence of embedded convection in WCBs reported in previous case studies (e.g., Neiman et al., 1993; Crespo and Posselt, 2016; Flaounas et al., 2016; Rasp et al., 2016; Oertel et al., 2019) and illustrates the large heterogeneity of the WCB cloud band. The online trajectories from the convection-permitting simulation, airborne radar observations, and satellite-based cloud top pressure qualitatively agree with respect to strength and location of embedded convection. The main conclusions of this first part are the following:

The consideration of detailed WCB ascent in the entire cyclone suggests that intense convective ascent with $\Delta p_{2h} < -600\,\mathrm{hPa}$ is primarily embedded near the cyclone center and at the cold front south of $45°\mathrm{N}$. In this region, the cloud top is heterogeneous and cloud tops locally exceed $200\,\mathrm{hPa}$. Moreover, convective WCB ascent with $\Delta p_{2h}$ values between $-350$ and $-400\,\mathrm{hPa}$ also occurs near the warm front and north of $45°\mathrm{N}$ and forms an integral part of the extended WCB cloud band. The variation of the intensity of WCB-embedded convection is directly reflected in the cloud structure. Airborne radar observations of stratiform and convective regions in the northern part of the WCB cloud band reveal the heterogeneous cloud structure associated with the WCB cloud band, and qualitatively reflect the differing WCB ascent rates and varying degrees of embedded convection. Qualitatively, the location and intensity of embedded convection in the radar observations agree with the online trajectories. While the stratiform region of the WCB cloud band coincides with slow WCB ascent with $\Delta p_{2h}$ values of approximately $-50\,\mathrm{hPa}$, the regions of embedded convection with vertically extended narrow plumes of enhanced radar reflectivity concur with $\Delta p_{2h}$ values below $-350$ and $-400\,\mathrm{hPa}$. This also underlines the ability of the online WCB trajectories to reasonably distinguish between faster and slower convective ascent.

In the second part, the detailed ascent behaviour of the WCB online trajectories is used to select two sub-categories of embedded convection: (i) moderate convective WCB trajectories with an embedded 400-hPa ascent time below 1 h, and (ii) intense convective WCB trajectories that perform their full $600\,\mathrm{hPa}$ ascent in less than 1 h. In this cyclone, the most intense embedded convection, with ascent rates exceeding $600\,\mathrm{hPa}$ in 30-60 min, was even faster than reported in previous case studies (e.g., Rasp et al., 2016; Oertel et al., 2019, 2020; Blanchard et al., 2020). In total, approximately 10% of all WCB trajectories associated with cyclone Sanchez perform such an intense convective ascent.

The systematic Lagrangian composite analysis for the intense and moderate convective WCB trajectories – and the comparison with WCB-embedded convection in cyclone Vladiana (Oertel et al., 2020) with an average convective ascent of $600\,\mathrm{hPa}$ in 3 h – highlights the direct influence of embedded convection on the cloud and precipitation structure (Table 1). The intense convective WCB ascent produces more intense surface precipitation (peak values of 15 minute accumulated surface precipitation of $> 6\,\mathrm{mm}$ compared to 3.9 mm for moderate convection and 4.5 mm for cyclone Vladiana) and larger amounts of graupel





during the ascent (peak values of $> 2.8 \, \mathrm{g \, kg^{-1}}$ compared to $>1.0 \, \mathrm{g \, kg^{-1}}$ for moderate convection and $>1.8 \, \mathrm{g \, kg^{-1}}$ for cyclone Vladiana).

Both convective WCB sub-categories ascend from a warm and moist region in the lower troposphere ahead of the cold front. The vertical structure of $\theta_e$ shows that moderate and intense convection both originate from a region of enhanced potential instability, but only the intense convective ascent in this study was consistently associated with higher values of CAPE prior to the onset of the convective ascent ($1000 \, \mathrm{J \, kg^{-1}}$ compared to $90 \, \mathrm{J \, kg^{-1}}$ for moderate convection and $290 \, \mathrm{J \, kg^{-1}}$ for cyclone Vladiana).

The composite PV structure in the upper troposphere highlights the complex effects of embedded convection on the upper-level flow and PV distribution (in contrast to its rather direct impact on the cloud and precipitation structure). In agreement with the previous case study of cyclone Vladiana (Oertel et al., 2020) and theoretical considerations (Chagnon and Gray, 2009; Harvey et al., 2020), both sub-categories of WCB-embedded convection form quasi-horizontal upper-level PV dipoles. However, we showed that the size, magnitude and orientation of the convectively generated PV dipoles – and hence their potential impact

on the large-scale upper-level flow – can differ substantially (Table 1). Compared to cyclone Vladiana, the convective ascent associated with the WCB of cyclone Sanchez is faster, but – despite the more intense convective ascent and associated latent heating – the PV dipole magnitude is lower, and its horizontal and vertical extent are smaller. Moreover, the orientation of the PV dipoles is reversed: While the negative PV pole was located west of the convective updraft near the upper-level jet in cyclone Vladiana, the negative PV pole in this study is located east of the convective updraft and away from the upper-level jet. These

differences in the PV dipole formation are directly attributed to the ambient vertical wind shear, which strongly modulates the PV dipole formation, and hence plays a key role for the impact of WCB-embedded convection on the upper-level flow.

This implies that the intensity of embedded convection alone is not a reliable measure for the effect of embedded convection on upper-level PV modification. Instead, the alignment of embedded convection and the vertical wind shear vector determine the characteristics of the PV dipole. The largest potential to produce coherent upper-level negative PV features near the jet arises

if persistent convective ascent is located close to a strong upper-level jet and aligned with the vertical wind shear vector (see Harvey et al., 2020). This emphasizes that the dynamical relevance of WCB-embedded convection does not only depend on the ascent strength (as for cloud and precipitation structure) but is strongly modulated by the synoptic situation, i.e., the location, geometry and alignment of embedded convection and the upper-level jet.

Despite the limited influence of embedded convection on the upper-level flow evolution in this study, we still hypothesize

that generally embedded convection in WCBs has the potential to modify the upper-level flow, because WCBs often ascend in the baroclinic environment of extratropical cyclones ahead of an upper-level jet (Harvey et al., 2020). Finally, this study highlights the important case-to-case variability of embedded convection in WCBs not only in terms of frequency and intensity of embedded convection but also in terms of its dynamical implications. A climatological investigation of embedded convection in WCBs using satellite observations and extended convection-permitting simulations of the extratropical storm track region

would allow for a detailed quantification of the relevance of embedded convection in WCBs.



*Data availability.* All data are available from the authors upon request.

*Author contributions.* AO performed the simulation and the data analysis, and prepared a first version of the paper. HK and MH provided the calibrated radar reflectivity data. All authors continuously discussed the results and contributed to the final manuscript.

*Competing interests.* The authors declare that they have no conflict of interest.

*Acknowledgements.* AO and MB acknowledge funding by the Swiss National Science Foundation (Project 165941), MP acknowledges funding from the European Research Council 485 (ERC) under the European Union's Horizon 2020 Research and Innovation program (project INTEXseas, grant agreement no. 787652), and HK acknowledges funding by the HALO Priority Program SPP 1294. The COSMO simulation was performed at the Swiss National Supercomputing Centre (CSCS), as part of the project sm08 (2017–2019). We thank the World Wide Lightning Location Network (http://wwlln.net), a collaboration among over 50 universities and institutions, for providing the

lightning location data. We appreciate the data from EUMETSAT used to visualize the WCB cloud band. Finally, we would like to thank all the NAWDEX colleagues for the successful measurement campaign.



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
