# Peer review of "Observations and simulation of intense convection embedded in a warm conveyor belt – how ambient vertical wind shear determines the dynamical impact"

_Weather and Climate Dynamics, 2020_

## Referee Comment (RC1) · Derek J. Posselt (Referee) · 4 Dec 2020

Review of Observations and simulation of intense convection embedded in a warm conveyor belt – how ambient vertical wind shear determines the dynamical impact

by

A. Oertel, M. Sprenger, H. Joos, M. Boettcher, H. Konow, M. Hagen, and H. Wernli

Summary:

[Figure]

This paper documents the observed occurrence of convective and stratiform features in an intense ETC that occurred during the NAWDEX field campaign. This storm was observed multiple times using airborne high altitude radar, and was also simulated using a convection permitting mesoscale model. The paper very effectively blends the observational and modeling analysis to diagnose the occurrence of convection in the storm, and presents a comprehensive analysis of the dynamic, thermodynamic, and cloud features. The authors find interesting differences between the occurrence and properties of moderate vs strong convection, and comparison vs an earlier anlyzed case indicates there is significant storm-to-storm variability.

This is a very well written and comprehensive analysis, and I found little to criticize. I have only a few questions and suggestions, and detail these below.

Questions / comments:

1. According to the online documentation, the MIRA-36 radar includes doppler velocity. I was curious as to what the observed updraft strengths in the radar observations were? If Doppler observations are available, it would be very interesting to see these plotted alongside of the reflectivity plots.

2. A very minor request - in Fig. 1 it would be helpful if there were text located above each column of sub-figures indicating the date/time of analysis.

3. The authors use observations from the WWLLN. I am curious as to whether the WWLLN observations extend north of 40 degrees latitude? If so, was lightning detected in the warm frontal region at any time during the storm development?

4. I thought it was interesting that the intense convection plotted in Fig. 6 appeared to occur within the warm front at earlier times, then shifted southward to along the cold front later (as the parent storm propagated northward). I wonder if, at later times, the strong convection to the south effectively stabilized the WCB air that later entered the region with moderate convection? Is it possible that, had there not been convection

along the CF, that the convection in the WF near the cyclone center might have been stronger? I am thinking of this from the perspective of a thermodynamic (or perhaps available convective available potential energy) budget. . .

—————————————————————

---

## Referee Comment (RC2) · Jeffrey Chagnon (Referee) · 16 Dec 2020

Summary:

This article presents a thorough and careful analysis of Lagrangian particles in the warm conveyor belt (WCB) of a cyclone that was simulated using the high-resolution COSMO model. Cyclone Sanchez was also the subject of an Intense Observing Period (IOP) during the NAWDEX field campaign. The study establishes that 1) the convective nature of the WCB in Cyclone Sanchez was simulated reasonably well in COSMO

compared to observations, 2) the intensity of convection varied according to the location along the cold front, and 3) only a weak horizontally-oriented potential vorticity dipole accompanied the convective trajectories. This third result, which is perhaps the most interesting outcome of the analysis, is hypothesized to be a consequence of the very weak vertical shear of the environment where ascent took place. A previous case study by Oertel et al. (2020) demonstrated a much larger amplitude PV structure in a cyclone where the environmental shear was larger.

The paper is well written, well organized, and very well presented. The analysis leverages a suite of tools for the analysis of trajectories that have time and again proved their value in revealing the detailed structure and dynamics of extratropical cyclones. I can find no major errors in the design, execution, and interpretation of the experiments. Overall, this work makes a valuable contribution to an ongoing area of research concerning the role of embedded convection in modifying the structure and evolution of an extratropical cyclone. I am therefore pleased to recommend publication. I have several general comments for the authors to consider, but I endorse publication in its current form.

General comments:

1. The relationship between the environmental shear and the PV dipole orientation/amplitude is consistent with theory. The authors have now demonstrated this in two cyclones (i.e., Sanchez and Vladiana). The differences between these two cases are quite remarkable. Given a sample size of only two, how robust and generalizable are these results? We know that the environmental shear where parcels ascended convectively was quite different in these two cases. How representative are these two storms of the bigger population of extratropical cyclones? What accounts for the difference in shear? What might we expect of storms in different parts of the world or in different months? What else besides the shear distinguishes these two storms and may contribute to the differences? I appreciate that the analysis of a single case is a significant labor, so the scope of this paper is appropriate, and the authors do allude to

this issue in the very last sentence of the paper. Nevertheless, the conclusions section could benefit from an expanded discussion.

2. The focus of the discussion around Figure 8 (which shows the composited PV structures) is on the horizontal orientation of PV dipoles relative to the trajectories. A vertically-oriented dipole structure is also evident in the sections from Cyclone Sanchez. The negative pole of the vertical dipole is of large amplitude [i.e., O(1 PVU)] and appears to have a broad horizontal extent near the tropopause level. Why isn't this vertical dipole structure and its consequences discussed? It is noteworthy that such a structure is not as evident in the case of Cyclone Vladania. What accounts for the difference?

3. Finally, I am curious about the relationship between trajectory location, PV tendencies, and the fidelity of compositing PV from trajectories. The trajectories are selected based on their ascent rate. It is therefore likely that trajectories are collocated to regions of maximum diabatic warming. The associated PV tendencies are therefore likely to be minimized along the trajectories (and maximized on the periphery). In compositing the PV structures associated with many trajectories, one could envision a significant degree of destructive interference. This interference might be most significant when trajectories are positioned next to one another across the PV dipole axis (i.e., stacked on top of one another for vertical dipoles; horizontally adjacent to one another in a direction perpendicular to the shear for horizontal dipoles). Have the authors considered this?

---

## Author Comment (AC1) · 13 Jan 2021

**Manuscript wcd-2020-49**

**Observations and simulation of intense convection embedded in a warm conveyor belt – how ambient vertical wind shear determines the dynamical impact**

Oertel, A., Sprenger, M., Joos, H., Boettcher, M., Konow, H., Hagen, M., and Wernli, H.

**Final author comments**

We would like to thank both reviewers, Derek J. Posselt and Jeffrey Chagnon, very much for their very positive, detailed and constructive feedback that helped to further improve the quality of this manuscript. In the following document, we address the questions and comments by the reviewers. Below are the detailed replies to the individual comments.

**1 Response to Derek J. Posselt**

**Comments to the author**

1. According to the online documentation, the MIRA-36 radar includes doppler velocity. I was curious as to what the observed updraft strengths in the radar observations were? If Doppler observations are available, it would be very interesting to see these plotted alongside of the reflectivity plots.

   **Reply** Thanks for this additional suggestion. Figure 1 in this document shows the Doppler velocity for the three flight segments shown in Figs. 2b,c and 3b in the manuscript. The Doppler velocity exhibits larger variability and locally reduced Doppler velocity (i.e., enhanced updrafts) in the regions which we identified as convective (Fig. 1b,c in this document) based on the radar reflectivity in the manuscript, while the stratiform regions display a rather homogeneous Doppler velocity with little along-flight variability (Fig. 1a in this document). We decided to not include the Doppler velocity in the manuscript for the following reasons: (i) The measurement of the Doppler velocity was affected by aircraft motion, and is also not provided in the official release of the calibrated data set (see Konow et al., 2019). The HALO cloud radar has a pointing direction wich is fixed perpendicular to the aircraft fusselage. This is in contrast to the systems of, e.g., Wyoming King-Air radar or the cloud radar flown with NCARs G5, their radars are always looking nadir. Doppler velocity (line-of-sight velocity) of the HALO radar is always affected by the attitude of the aircraft, mainly the pitch of 2–3°, and the aircraft velocity. The Doppler velocity can be corrected for attitude and aircraft speed, which is done for Fig. 1. However, there is still some unresolved noise in the attitude data which is seen in Fig. 1 as fluctuations with thin vertical stripes. (ii) It is difficult to obtain an accurate estimate of the updraft strengths of embedded convection, as this would require the exact horizontal wind field and the fall velocity of hydrometeors, and the estimation of hydrometeor fall velocity is still associated with large uncertainties.

[Figure]

Figure 1: HALO Doppler velocity $v_D$ (in $\mathrm{m\,s^{-1}}$) along the flight segments shown in **(a)** Fig. 2b, **(b)** Fig. 2c, and **(c)** Fig. 3b in the manuscript, including WCB mask from ECMWF WCB trajectories (dark blue contour and hatching), the $0°$C isotherm (yellow) and the $2\,\mathrm{PVU}$ contour (red). The green line in **(b)** at 13:40 UTC and **(c)** at 14:50 UTC marks the aircraft turning points. Positive values of $v_D$ denote downward motion of hydrometeors away from the aircraft.

2. A very minor request - in Fig. 1 it would be helpful if there were text located above each column of sub-figures indicating the date/time of analysis.
**Reply** Thanks for this helpful comment, the date/time was added to the figure.

3. The authors use observations from the WWLLN. I am curious as to whether the WWLLN observations extend north of 40 degrees latitude? If so, was lightning detected in the warm frontal region at any time during the storm development?
**Reply** The WWLLN observations extend north of 40°N, however, generally fewer lightning strokes are detected in higher latitudes in the North Atlantic sector. For the presented case study, the lightning observations are mostly restricted to the cold front and cyclone center. We find one timestep (Fig. 2 in this document), where a lightning stroke was detected ahead of the warm front north of 40°N. This rare occurrence of lightning at the warm front is qualitatively in line with the COSMO simulation, which shows more intense embedded convection at the cold front and in the cyclone center.
However, please note that the WWLLN has a comparatively low detection efficiency of only 10% for moderately strong and 35% for very strong currents (Abarca et al., 2010). Hence, the WWLLN observations likely underestimate the occurrence of lightning, in particular at the warm front, where convective activity is weaker. Further details and applications of the WWLLN data can be found in, e.g., Jacobson et al. (2006), Abarca et al. (2010, 2011), and McTaggart-Cowan (2010).

4. I thought it was interesting that the intense convection plotted in Fig. 6 appeared to occur within the warm front at earlier times, then shifted southward to along the cold front later (as the parent storm propagated northward). I wonder if, at later times, the strong convection to the south effectively stabilized the WCB air that later entered the region with moderate convection? Is it possible that, had there not been convection along the CF, that the convection in the WF near the cyclone center might have been stronger? I am thinking of this from the perspective of a thermodynamic (or perhaps available convective available potential energy) budget.
**Reply** This is a good question, which will remain speculative. In terms of thermodynamic equilibrium it makes sense that once convection has set in at the CF, the atmospheric column is stabilized (i.e., release of CAPE) and moist and warm air from the lower troposphere has been transported into the upper troposphere in the convective updrafts. Beside increasing the static stability, this also transports moisture from the lower into the upper troposphere. Thus, less moisture is available for subsequent cross-isentropic WCB ascent and embedded convection, which we believe will influence the ensuing WCB ascent behaviour. We hypothesize that a potential suppression of convective activity ahead of the CF, could indeed lead to more intense convection in the later stage of the cyclone, e.g., near the cyclone center or the WF, because (i) potential instability has not been fully removed, and (ii) more low-level moist and warm air is still present. However, this development also strongly depends on the overall WCB ascent, i.e., how much mass transport and stabilization of the troposphere has been performed by slantwise and gradual WCB ascent. Hence, we have no final answer to how a potential suppression of convective activity ahead of the CF might influence convective activity at a later stage near the cyclone center and at the WF.

[Figure]

Figure 2: Equivalent potential temperature at $850\,\text{hPa}$ (THE, in K) and lightning obser-vations (yellow crosses) from WWLLN at 05 UTC 10 Oct 2016 to illustrate the detection of a single lightning stroke at the warm front.

**2 Response to Jeffrey Chagnon**

**Comments to the author**

1. The relationship between the environmental shear and the PV dipole orientation / amplitude is consistent with theory. The authors have now demonstrated this in two cyclones (i.e., Sanchez and Vladiana). The differences between these two cases are quite remarkable. Given a sample size of only two, how robust and generalizable are these results? We know that the environmental shear where parcels ascended convectively was quite different in these two cases. How representative are these two storms of the bigger population of extratropical cyclones? What accounts for the difference in shear? What might we expect of storms in different parts of the world or in different months? What else besides the shear distinguishes these two storms and may contribute to the differences? I appreciate that the analysis of a single case is a significant labor, so the scope of this paper is appropriate, and the authors do allude to this issue in the very last sentence of the paper. Nevertheless, the conclusions section could benefit from an expanded discussion.
**Reply** We agree that the generalization of these questions would be highly interesting, and more research on this topic would be beneficial. We added a more elaborate discussion in the "conclusions and discussions" section in the manuscript.

2. The focus of the discussion around Figure 8 (which shows the composited PV structures) is on the horizontal orientation of PV dipoles relative to the trajectories. A vertically-oriented dipole structure is also evident in the sections from Cyclone Sanchez. The negative pole of the vertical dipole is of large amplitude [i.e., O(1 PVU)] and appears to have a broad horizontal extent near the tropopause level. Why isn't this vertical dipole structure and its consequences discussed? It is note worthy that such a structure is not as evident in the case of Cyclone Vladania. What accounts for the difference?
**Reply** We focused on the horizontal PV dipole structure at 7 km height, as the signal is substantially clearer, and an eloborate discussion of this negative PV feature at 10 km height might deviate the reader from our main message. We here try to explain the presence of this negative PV air in the upper troposphere, which is related to the complex flow situation and turning of the wind shear vector with height. In theory, if uni-directional wind shear were present, one would expect a PV dipole that is tilted in the vertical, whereby the degree of the tilt away from the vertical depends on the strength of the wind shear and the scale of the system (Chagnon and Gray, 2009). In the case of cyclone Sanchez, however, the wind shear turns with height, in particular between 7-9 km height (Fig. 9b in the manuscript). Hence, in this region also the PV dipole orientation turns with height, which is why it is not visible in the east-west orientated vertical cross-section. At 10 km height (approx. 350 K), the wind shear vector points to the north-west, hence, negative PV forms to the left of the wind shear vector (Fig. 3 in this document; see also Oertel et al., 2020). This negative PV feature is slightly shifted towards the center of the convective updrafts, which is why it appears to be "sitting on top of the updrafts" as a vertical dipole in Fig. 8e in the manuscript. Overall, the upper-tropospheric

[Figure]

Figure 3: Horizontal cross-section composite of PV (colors, in PVU) and wind speed (grey arrows, in $m\,s^{-1}$) at $335\,K$ for intense convective WCB trajectories in cyclone Sanchez 15 minutes after the start of the fastest 400-hPa ascent (i.e., as Fig. 8b in the manuscript but at $335\,K$). The green arrow shows the vertical wind shear vector between 9 and $10.5\,km$ height at the start of the WCB trajectory ascent. The axes' dimensions denote the distance from the WCB air parcel locations marked as '×' (in km).

PV signal is dominated by very low (but not negative) PV values resulting from PV reduction in an environment with larger-scale diabatic heating in the middle troposphere, where the vertical component of the PV tendency equation is important. As a side note, the vertical component of the PV tendency equation alone can only decrease PV values, but cannot form negative PV values (Harvey et al., 2020). This PV structure is not present in the Vladiana case study, because the wind shear does not rotate with height (Fig. 9b in the manuscript), and hence forms one coherent PV dipole (Fig. 8f in the manuscript).

3. Finally, I am curious about the relationship between trajectory location, PV tendencies, and the fidelity of compositing PV from trajectories. The trajectories are selected based on their ascent rate. It is therefore likely that trajectories are collocated to regions of maximum diabatic warming. The associated PV tendencies are therefore likely to be minimized along the trajectories (and maximized on the periphery). In compositing the PV structures associated with many trajectories, one could envision a significant degree of destructive interference. This interference might be most significant when trajectories are positioned next to one another across the PV dipole axis (i.e., stacked on top of one another for vertical dipoles; horizontally adjacent to one another in a direction perpendicular to the shear for horizontal dipoles). Have the authors considered this?

Reply To account for the vertical position of trajectories, we centered the trajectories such that they display a similar ascent behaviour (Fig. 5 in the manuscript) and for each time step we consider only trajectories that are located at approximately the same height (hence, in a comparable ascent state), which maximizes the signal. Concerning horizontal adjacency, previous studies found that often convective updrafts (Oertel et al. 2020) or narrow heating regions (Harvey et al. 2020) form narrow banded structures parallel to the cold front and upper-level jet system, which is in line with observations of front-parallel banded precipitation (e.g., Bennetts and Hoskins, 1979, Siedersleben et al., 2016, Jeyaratnam et al., 2016). Hence, the compositing technique would provide clear dipoles (Fig. 4a in this document), although on very small-scales some destructive interference is probably present (this also to some extent results in the smaller-amplitude PV dipole in the composite analysis (order of $\pm 1\,\mathrm{PVU}$) compared to the amplitudes of instantaneous PV dipoles (order of $\pm 10\,\mathrm{PVU}$)). For the presented case study, Fig. 6a in the manuscript shows that convective trajectories are also located in rather narrow quasi-parallel bands. Assuming a strong upper-level jet were to be located in this region, this would also result in a clear PV dipole composite structure. In contrast, if the diabatic heating were to be arranged in bands perpendicular to the shear vector (i.e., trajectories are horizontally adjacent to one another perpendicular to the shear, Fig. 4b in this document), no distinct PV dipoles would form, and no signal would be present in the composite PV field. However, this is not a problem of the trajectory compositing technique per se, because in such a situation also in a Eulerian perspective no distinct PV dipole structure would form as no strong horizontal heating gradients are aligned with the horizontal vorticity vector (neglecting a potentially smaller signal at the "edges" of the elongated heating region). Generally, compositing techniques will always include some destructive interferences, and we see no substantial difference between compositing relative to selected trajectory positions or, e.g., maximum updrafts (Weijenborg et al., 2017). However, using trajectory-relative composites additionally provides the advantage to observe the life cycle (i.e., characteristics prior or after convective ascent). Specifically for our analysis, the trajectory-based composite analysis also ensures that trajectories perform a deep WCB-like ascent.

[Figure]

Figure 4: Sketch of hypothetical spatial arrangements of embedded convective activity: Convective cloud bands arranged **(a)** parallel, and **(b)** perpendicular to the front-jet system. The arrow indicates the direction of the wind shear vector, blue and red colours are negative and positive PV anomalies, and black crosses are individual convective trajectories.

**References**

Abarca, S. F., K. L. Corbosiero, and T. J. Galarneau Jr., 2010: An evaluation of the Worldwide Lightning Location Network (WWLLN) using the National Lightning Detection Network (NLDN) as ground truth. J. Geophys. Res., 115, D18 206, https://doi:10.1029/2009JD013411.

Abarca, S. F., K. L. Corbosiero, and D. Vollaro, 2011: The World Wide Lightning Location Network and convective activity in tropical cyclones. Mon. Wea. Rev., 139, 175–91, https://doi:10.1175/2010MWR3383.1

Bennetts, D. A. and Hoskins, B. J., 1979: Conditional symmetric instability - a possible explanation for frontal rainbands, Q. J. Roy. Meteorol. Soc., 105, 945–962, https://doi.org/10.1002/qj.49710544615.

Chagnon, J. M. and Gray, S. L., 2009: Horizontal potential vorticity dipoles on the convective storm scale, Q. J. R. Meteorol. Soc., 135, 1392–1408, https://doi.org/10.1002/qj.468.

Harvey, B., Methven, J., Sanchez, C., and Schäfler, A., 2020: Diabatic generation of negative potential vorticity and its impact on the North Atlantic jet stream, Q. J. R. Meteorol. Soc., 146, 1477–1497, https://doi.org/10.1002/qj.3747.

Jacobson, A., R. H. Holzworth, J. Harlin, R. L. Dowden, and E. Lay, 2006: Performance assessment of the World Wide Lightning Location Network (WWLLN), using the Los Alamos Sferic Array (LASA) array as ground-truth. J. Atmos. Oceanic Tech., 23, 1082–92, https://doi:10.1175/JTECH1902.1.

Jeyaratnam, J., Booth, J. F., Naud, C. M., Luo, Z. J., and Homeyer, C. R., 2020: Upright convection in extratropical cyclones: A survey using ground-based radar data over the United States, Geophys.Res. Lett., 47, e2019GL086 620, https://doi.org/10.1029/2019GL086620.

Konow, H., Jacob, M., Ament, F., Crewell, S., Ewald, F., Hagen, M., Hirsch, L., Jansen, F., Mech, M., and Stevens, B., 2019: A unified data set of airborne cloud remote sensing using the HALO Microwave Package (HAMP), Earth Syst. Sci. Data, 11, 921–934, https://doi.org/10.5194/essd-11-921-2019.

McTaggart-Cowan, R., 2010: Development and tropical transition of an Alpine lee cyclone. Part I: Case analysis and evaluation of numerical guidance. Mon. Wea. Rev., 138, 2281–2307, https://doi:10.1175/2009MWR3147.1.

Oertel, A., Boettcher, M., Joos, H., Sprenger, M., and Wernli, H., 2020: Potential vorticity structure of embedded convection in a warm conveyor belt and its relevance for large-scale dynamics, Weather Clim. Dynam., 1, 127–153, https://doi.org/10.5194/wcd-1-127-2020.

Siedersleben, S. K. and Gohm, A., 2016: The missing link between terrain-induced

potential vorticity banners and banded convection, Mon. Weather Rev., 144, 4063–4080, https://doi.org/10.1175/MWR-D-16-0042.1.

Weijenborg, C., Chagnon, J. M., Friederichs, P., Gray, S. L., and Hense, A., 2017: Coherent evolution of potential vorticity anomalies associated with deep moist convection, Q. J. R. Meteorol. Soc., 143, 1254–1267, https://doi.org/10.1002/qj.3000.